# Large Language Model Reasoning Failures

**Peiyang Song** *†                                                    *psong@caltech.edu*
*California Institute of Technology, Stanford University*

**Pengrui Han** *                                                       *barryhan@carleton.edu*
*Carleton College*

**Noah Goodman**                                                        *ngoodman@stanford.edu*
*Stanford University*

**Reviewed on OpenReview:** https://openreview.net/forum?id=vnX1WHMNmz

## Abstract

Large Language Models (LLMs) have exhibited remarkable reasoning capabilities, achieving impressive results across a wide range of tasks. Despite these advances, significant reasoning failures persist, occurring even in seemingly simple scenarios. To systematically understand and address these shortcomings, we present the ***first comprehensive survey dedicated to reasoning failures in LLMs***. We introduce a novel categorization framework that distinguishes ***reasoning*** into embodied and non-embodied types, with the latter further subdivided into informal (intuitive) and formal (logical) reasoning. In parallel, we classify reasoning ***failures*** along a complementary axis into three types: fundamental failures intrinsic to LLM architectures that broadly affect downstream tasks; application-specific limitations that manifest in particular domains; and robustness issues characterized by inconsistent performance across minor variations. For each reasoning failure, we provide a clear definition, analyze existing studies, explore root causes, and present mitigation strategies. By unifying fragmented research efforts, our survey provides a structured perspective on systemic weaknesses in LLM reasoning, offering valuable insights and guiding future research towards building stronger, more reliable, and robust reasoning capabilities. We additionally release a comprehensive collection of research works on LLM reasoning failures, as a GitHub repository at https://github.com/Peiyang-Song/Awesome-LLM-Reasoning-Failures, to provide an easy entry point to this area.

## 1 Introduction

> *"Failure is success if we learn from it." – Malcolm Forbes*

With the rise of powerful architectures (Vaswani et al., 2023; Jiang et al., 2024a; Gu & Dao, 2024; Hasani et al., 2020), efficient algorithms (Hu et al., 2021; Zhao et al., 2024b; Gretsch et al., 2024; 2025; Dao et al., 2022), and massive data (Cai et al., 2024; Raffel et al., 2020; Gao et al., 2020), Large Language Models (LLMs) have recently shown significant success across diverse domains. These range from traditional linguistic tasks such as machine translation (Zhu et al., 2024b; Tang et al., 2024), to mathematical (Shao et al., 2024; Yang et al., 2023a; 2024a) and even scientific (Zhang et al., 2024b; Wang et al., 2023b; Brodeur et al., 2024) discoveries. Among these achievements, reasoning as an emergent capability of LLMs (Wei et al., 2022a) has attracted particular interest (Huang & Chang, 2023; Yu et al., 2023b; Qiao et al., 2023).

LLMs have set impressive records in reasoning (Wu et al., 2025a; Kıcıman et al., 2024; Plaat et al., 2024), though it remains controversial whether LLMs really leverage a human-like reasoning procedure when

---

*Equal contribution.
†Work done while Peiyang Song was a visiting researcher at Stanford University.

attempting these tasks (Jiang et al., 2024b; Fedorenko et al., 2024; Amirizaniani et al., 2024b; Zhang et al., 2022). This survey does not aim to settle this hot debate; rather we focus on an important area of study in LLM reasoning that has long been overlooked – LLM reasoning failures.

Extensive psychological research (Cannon & Edmondson, 2005; Maxwell, 2007; Coelho & McClure, 2004) underscores the importance of identifying and learning from failures in human development[1]. Given that AI systems have historically drawn inspiration from human cognition (Schmidgall et al., 2023; Xu & Poo, 2023; Woźniak et al., 2020), we believe the same principle of learning from failures could similarly benefit the study of LLMs, since such failures can usually be traced back to fundamental elements and bring valuable insights to ultimate improvements (Dreyfus, 1992; Karl et al., 2024; An et al., 2024).

Despite some existing works that prospectively realized this importance and investigated LLM reasoning failures on a case-by-case basis (Williams & Huckle, 2024; Tie et al., 2024; Helwe et al., 2021; Borji, 2023), the topic remains fragmented, and underexplored as a unified research area. This fragmentation limits broader understanding, which is however a prerequisite for common patterns to be noticed, and thereby meaningful lessons to be derived. To close this gap, we present the first comprehensive survey dedicated to unifying studies on LLM reasoning failures. We identify meaningful patterns across failures, analyze underlying causes, and discuss potential mitigation strategies. We hope this work not only organizes the field but also stimulate further research and increased attention, toward improving the robustness and reliability of LLM reasoning. We additionally make public a comprehensive collection of research works on LLM reasoning failures, as a GitHub repository at https://github.com/Peiyang-Song/Awesome-LLM-Reasoning-Failures. This collection will be continuously updated as this area advances.

## 2 Definition and Formulation

### 2.1 Fundamentals of Reasoning

Human reasoning broadly refers to the ability to draw conclusions and make decisions based on available knowledge (Lohman & Lakin, 2011; Ribeiro et al., 2020). Within cognitive science and philosophy, reasoning has been studied through various frameworks. To systematically survey reasoning failures in LLMs, we propose a comprehensive taxonomy distinguishing reasoning along two primary axes: ***embodied*** versus ***non-embodied***, with the latter further subdivided into ***informal*** and ***formal*** reasoning.

**Non-embodied reasoning.**   Non-embodied reasoning comprises cognitive processes not requiring physical interaction with environments. Within this category, *informal reasoning* encompasses intuitive judgments driven by inherent biases and heuristics, common in everyday decision-making and social activities (Piaget, 1952; Vygotsky, 1978; Kail, 1990). By contrast, *formal reasoning* involves explicit, rule-based manipulation of symbols, grounded in logic, mathematics, code, etc. (Copi et al., 2016; Mendelson, 2009; Liu et al., 2023b).

**Embodied reasoning.**   Embodied reasoning depends on physical interaction with environments, fundamentally relying on spatial intelligence and real-time feedback (Shapiro, 2019; Barsalou, 2008). This includes predicting and interpreting physical dynamics, and performing goal-directed behaviors constrained by real-world physical laws (Huang et al., 2022b; Lee-Cultura & Giannakos, 2020).

### 2.2 LLM Reasoning Failures & Common Research Practice

Despite advances in interpretability research (Dwivedi et al., 2023; Li et al., 2024e), LLMs remain largely *black-box* systems (Luo & Specia, 2024), reflecting the inherent complexity of human cognition they emulate (Castelvecchi, 2016). As such, reasoning abilities are typically assessed behaviorally by examining model outputs on carefully designed prompts and tasks (Ribeiro et al., 2020). We define ***LLM reasoning failures*** as cases where model responses significantly diverge from expected logical coherence, contextual relevance, or factual correctness. Failures can manifest in two broad ways. The first type is straightforward poor performance — the model fails decisively on a task, exposing clear deficiencies. The second, subtler type involves apparently

---

[1]In fact, this theory has been confirmed even more broadly, in non-human animals (Spence, 1936).

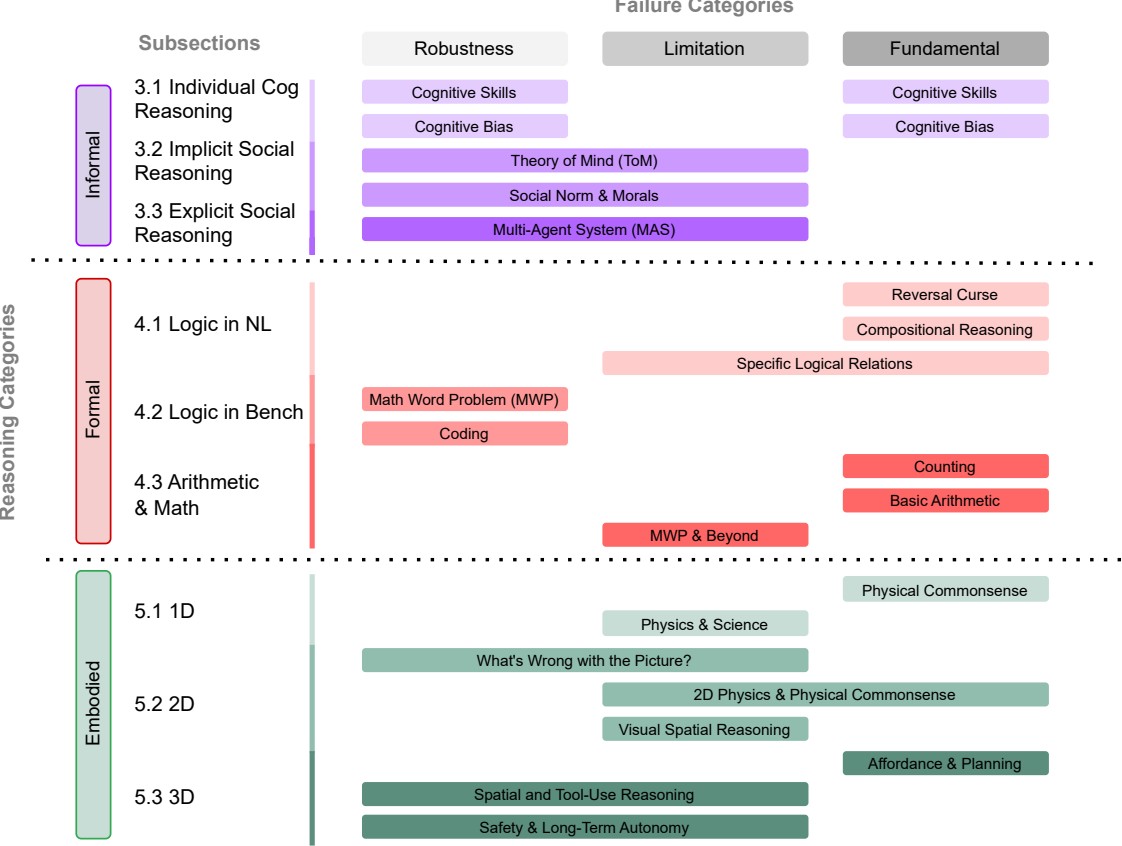

Figure 1: **A Taxonomy of LLM Reasoning Failures.** We adopt a nuanced 2-axis structure (reasoning type × failure type), with each row representing a reasoning category and each column a failure category. A more detailed explanation is presented in Section 2.

adequate performance that is in fact unstable, indicating a ***robustness*** issue that reveals hidden vulnerabilities. The former category – straightforward failure – can be sub-divided into two, based on scope and nature. ***Fundamental*** failures are usually intrinsic to LLM architectures, manifesting broadly and universally across diverse downstream tasks. In contrast, application-specific ***limitations*** reflect shortcomings tied to particular domains of importance, where models underperform despite human expectations of competence. Together, these taxonomies — for reasoning and for failures — offer a comprehensive and mutually consistent framework. Figure 1 uses this framework to visualize a clear organization of topics in this survey.

Current research in this space typically begins with ***simple, intuitive tests*** that reveal glaring reasoning failures. These initial observations motivate ***larger-scale systematic evaluations***, to confirm the generality and impact of identified failure modes. By explicitly defining and categorizing LLM reasoning failures according to our framework, this survey unifies fragmented research findings, highlights shared patterns, and directs focused efforts toward understanding and mitigating critical reasoning weaknesses. To help visualize the failure cases, we provide a few most representative examples for each of the failure case presented in this survey. The examples can be found in Appendix E.

# 3 Reasoning Informally in Intuitive Applications

Humans naturally develop the capacity for informal reasoning early in life, relying on intuitive judgments shaped by cognitive processes and social experiences. Though often taken for granted, this forms the foundation of human reasoning and decision-making. In this section, we focus on failures exhibited by LLMs in such informal reasoning. We begin by examining reasoning failures in core cognitive abilities reflected in individual LLM behaviors; then explore those exposed in social contexts, expressed implicitly or explicitly.

## 3.1 Individual Cognitive Reasoning

Many reasoning failures exhibited by LLMs can be traced back to core human cognitive phenomena (Han et al., 2024b; Gong et al., 2024; Galatzer-Levy et al., 2024; Suri et al., 2024). These failures arise either because LLMs lack certain fundamental cognitive abilities possessed by humans – leading to errors that humans typically avoid (Han et al., 2024b) – or because LLMs replicate human-like cognitive biases and heuristics, resulting in analogous mistakes (Suri et al., 2024; Lampinen et al., 2024). In both cases, these failures relate closely to **well-documented human cognitive phenomena** and psychological evidence.

**Fundamental Cognitive Skills.** Humans naturally possess a set of fundamental cognitive skills indispensable for reasoning. LLMs demonstrate systematic failures due to deficiencies in these areas. A prominent example is the set of **core executive functions** – working memory (Baddeley, 2020), inhibitory control (Diamond, 2013; Williams et al., 1999), and cognitive flexibility (Canas et al., 2006) – essential in human reasoning (Diamond, 2013). **Working memory** is the capacity to hold and manipulate information over short periods. LLMs' limited working memory leads to failures when task demands exceed their capacity (Gong et al., 2024; Zhang et al., 2024a; Gong & Zhang, 2024; Upadhayay et al., 2025; Huang et al., 2025a). In particular, LLMs suffer from "proactive interference" to a much larger extent than humans, where earlier information significantly disrupts retrieval of newer updates (Wang & Sun, 2025). **Inhibitory control** – the ability to suppress impulsive or default responses when contexts demand – is also weak in LLMs, with them often sticking to previously learned patterns even when contexts shift (Han et al., 2024b; Patel et al., 2025). Lastly, **cognitive flexibility**, the skill of adapting to new rules or switching tasks efficiently, remains a challenge, especially in rapid task switching and adaptation to new instructions (Kennedy & Nowak, 2024).

Another key aspect is **abstract reasoning** (Guinungco & Roman, 2020), the cognitive ability to recognize patterns and relationships in intangible concepts. Even advanced LLMs struggle with abstract reasoning tasks, such as inferring underlying rules from limited examples, understanding implicit conceptual relationships, and reliably handling symbolic or temporal abstractions (Xu et al., 2023c; Gendron et al., 2023; Galatzer-Levy et al., 2024; Saxena et al., 2025).

These phenomena are **fundamental** reasoning failures that stem from intrinsic limitations of LLM architectures and training dynamics, and often manifest as robustness vulnerabilities across a wide range of tasks. Recent work attributes these failures to the underlying self-attention mechanism's dispersal of focus under complex tasks (Gong & Zhang, 2024; Patel et al., 2025), and to the next token prediction training objective, which prioritizes statistical pattern completion over deliberate reasoning (Han et al., 2024b; Enström et al., 2024). Some also point out that unlike humans – who develop fundamental cognitive functions through embodied, goal-driven interactions with the physical and social world (Pearce & Miller, 2025; Rodríguez, 2022; Jin et al., 2018) – LLMs learn passively from text alone, lacking grounding and experiential feedback to support the development. Efforts to enhance these skills correspondingly include advanced prompting like Chain-of-Thought (CoT) (Wei et al., 2022b), retrieval augmentation (Xu et al., 2023b), fine-tuning with deliberately injected interference (Li et al., 2022), multimodality (Hao et al., 2025), and architectural innovations to mimic human attention mechanisms (Wu et al., 2024d).

**Cognitive Biases.** Cognitive biases – systematic deviations from rational judgment – are well-studied in human reasoning (Tversky & Kahneman, 1974; 1981). They arise from mental shortcuts, limited cognitive resources, or contextual influences, often leading to predictable errors. LLMs exhibit similar biases that systematically affect their reasoning across diverse tasks (Hagendorff, 2023; Bubeck et al., 2023). Since

these biases are deeply ingrained from training data and model architecture, they permeate a wide range of downstream applications, necessitating careful identification and mitigation.

In humans, these biases become evident only when information is presented and their responses observed – similarly, in LLMs, cognitive biases manifest also through the processing of information. Here lie two interrelated factors: **the content of information** and **the presentation of that information**. Regarding content, LLMs struggle more with abstract or unfamiliar topics – a phenomenon known as the "content effect" (Lampinen et al., 2024) – and tend to favor information that aligns with prior context or assumptions, reflecting human-like confirmation bias (O'Leary, 2025b; Shi et al., 2024; Malberg et al., 2024; Wan et al., 2025b; Zhu et al., 2024c). Social cognitive biases also influence LLM outputs, including group attribution bias (Hamilton & Gifford, 1976; Allison & Messick, 1985; Raj et al., 2025) and negativity bias (Rozin & Royzman, 2001), which prioritize popular content (Echterhoff et al., 2024; Lichtenberg et al., 2024; Jiang et al., 2025a) and negative inputs (Yu et al., 2024c; Malberg et al., 2024) respectively.

Equally important is how the same content is presented. LLMs are highly sensitive to the order in which information is given, exhibiting order bias (Koo et al., 2023; Pezeshkpour & Hruschka, 2023; Jayaram et al., 2024; Guan et al., 2025; Cobbina & Zhou, 2025), and show anchoring bias (Lieder et al., 2018; Rastogi et al., 2022), where early inputs disproportionately shape their reasoning (Lou & Sun, 2024; O'Leary, 2025a; Huang et al., 2025e; Wang et al., 2025b). Framing effects further influence outputs: logically equivalent but differently phrased prompts can lead to different results (Jones & Steinhardt, 2022; Suri et al., 2024; Nguyen, 2024; Lior et al., 2025; Robinson & Burden, 2025; Shafiei et al., 2025). Additionally, factors like narrative perspective (e.g., first-person vs. third-person) (Cohn et al., 2024; Lin et al., 2024b), prompt length or verbosity (Koo et al., 2023; Saito et al., 2023), and irrelevant or distracting information (Shi et al., 2023) further derail logical reasoning.

Cognitive biases constitute **fundamental** reasoning failures rooted in LLM training paradigms and architectures, and they manifest as **robustness** vulnerabilities across a wide range of downstream applications. The root causes of these cognitive biases in LLMs are threefold. First, biases are *inherited from the pre-training data*, where the linguistic patterns in human languages reflect cognitive errors (Itzhak et al., 2025). Second, architectural features of the model – such as the Transformer's causal masking – introduce predispositions toward order-based biases independent of data (Wu et al., 2025b; Dufter et al., 2022). Third, *alignment* processes like Reinforcement Learning from Human Feedback (RLHF) amplify biases by aligning model behavior with human raters who are themselves biased (Sumita et al., 2025; Perez et al., 2023).

Mitigation strategies fall into three categories. *Data-centric* approaches focus on curating training data to reduce biased content (Sun et al., 2025a; Schmidgall et al., 2024; Han et al., 2024a). *In-processing* techniques, such as adversarial training, aim to prevent biased associations during model learning (Yang et al., 2023b; Cantini et al., 2024). Lastly, *post-processing* methods leverage prompt engineering or output filtering to steer model responses after training (Sumita et al., 2025; Lin & Ng, 2023). In this category, indirect methods like inducing specific model personalities have also shown promise in modulating biases (Shi et al., 2024; He & Liu, 2025). Nonetheless, even when mitigated in one context, cognitive biases often re-emerge when contexts shift. The diverse and penetrative nature of cognitive biases makes them difficult to be fully eliminated.

### 3.2 Implicit Social Reasoning

Certain cognitive reasoning failures manifest only within social contexts. We define **implicit social reasoning** as an individual model's capacity to internally infer and reason about (1) others' mental states (e.g., beliefs, emotions, intentions) and (2) shared social norms *without requiring direct interaction.*

**Theory of Mind (ToM).** ToM is the cognitive ability to attribute mental states – beliefs, intentions, emotions – to oneself and others, and to understand that others' mental states may differ from one's own (Frith & Frith, 2005). ToM enables humans to interpret behaviors, predict actions, and navigate complex interpersonal interactions, central in social reasoning. Typically emerging in early childhood with milestones like passing false belief tasks (understand that others' beliefs may be incorrect or different) (Wimmer & Perner, 1983), ToM has been a central focus in human psychology and cognitive science.

Under this inspiration, recent research evaluates the ToM capacity of LLMs, to gauge their ability to engage in social reasoning. Early studies focused on classic ToM tasks, such as false-belief (van Duijn et al., 2023; Kim et al., 2023), perspective-taking (infer what another individual perceives) (Sap et al., 2022), and unexpected content tasks (predicting what others would believe is inside a mislabeled unopened container) (Pi et al., 2024). Surprisingly, even advanced models such as GPT-4 struggle with these tasks trivial for human children. Moreover, minor modifications in task phrasing lead to drastic drops in performance, showing LLM ToM reasoning is unstable (Ullman, 2023; Kosinski, 2023; Pi et al., 2024; Shapira et al., 2023).

While there has been clear progress from early models like GPT-3 – which largely failed at ToM tasks – to newer models such as GPT-4o and reasoning models like o1-mini, which can solve many standard ToM tests, their underlying reasoning remains brittle under simple perturbations (Gu et al., 2024; Zhou et al., 2023d). Also, LLMs still struggle with higher-order, more complex aspects of ToM, such as predicting others' behaviors, making appropriate moral or social judgments, and translating this understanding into coherent actions (He et al., 2023; Gu et al., 2024; Marchetti et al., 2025; Amirizaniani et al., 2024a; Strachan et al., 2024). Particularly, on dynamic, conversational benchmarks (Xiao et al., 2025; Kim et al., 2023), even state-of-the-art models fail to demonstrate consistent ToM capabilities and perform significantly worse than humans. Furthermore, current models exhibit deficits in *emotional* reasoning. This includes difficulties in emotional intelligence (EI) (Sabour et al., 2024; Hu et al., 2025; Amirizaniani et al., 2024b; Vzorinab et al., 2024), susceptibility to affective bias (Chochlakis et al., 2024), and limited understanding of cultural variations in emotional expression and interpretation (Havaldar et al., 2023).

While prompting techniques like CoT offer some improvements (Gandhi et al., 2024), fundamental gaps remain due to deeper limitations from the LLM architecture, training paradigms, and a lack of embodied cognition (Strachan et al., 2024; Sclar et al., 2023). Failures in this domain constitute important application-specific **limitations**, and because ToM underlies many socially grounded tasks, such failures often result in significant **robustness** vulnerabilities. Given ToM's centrality to social reasoning, future work should move beyond prompting, to probe deeper root causes and general mitigation.

**Social Norms and Moral Values.** LLMs also struggle with reasoning about social norms, moral values, and ethical principles that govern human behavior. Unlike humans, who develop moral and ethical reasoning through experience, LLMs, trained purely on text, often exhibit inconsistent and unreliable social, moral, and ethical reasoning (Ji et al., 2024; Jain et al., 2024b).

One key limitation is that LLMs cannot reason and apply *moral values* (Ji et al., 2024) and *social norms* (Jain et al., 2024b) consistently. They often produce contradictory ethical judgments or varied moral reasoning performance when questions are slightly reworded (Bonagiri et al., 2024), generalized (Tanmay et al., 2023), or presented in a different language (Agarwal et al., 2024). Fine-tuning further exacerbates these inconsistencies, leading to sometimes prioritizing task-specific optimization over ethical coherence (Yu et al., 2024a).

Beyond *inconsistencies*, LLMs show notable *disparities* compared to humans in reasoning with social norms and moral values. These models fail significantly in understanding real-world social norms (Rezaei et al., 2025), aligning with human moral judgments (Garcia et al., 2024; Takemoto, 2024), and adapting to cultural differences (Jiang et al., 2025b). Without consistent and reliable moral reasoning, LLMs are not fully ready for real-world decision-making involving ethical considerations.

These inconsistencies and disparities constitute important application-specific **limitations** for safety, privacy, sensitivity, and morality-related tasks, and such failures often create severe **robustness vulnerabilities**, including susceptibility to jailbreaks and other forms of manipulation. Many argue that these failures stem from a fundamental absence of robust, internalized representations of ethical principles, normative frameworks, and moral intentionality (Chakraborty et al., 2025; Wang et al., 2025a; Pock et al., 2023; Almeida et al., 2024). Although training procedures such as RLHF and instruction fine-tuning introduce alignment signals, they often operate superficially and fail to produce coherent moral behavior in complex contexts (Dahlgren Lindström et al., 2025; Wang et al., 2025a; Barnhart et al., 2025; Han et al., 2025). Current efforts to address these limitations mainly include prompt-based interventions (Chakraborty et al., 2025; Ma et al., 2023), internal activation steering (Tlaie, 2024; Turner et al., 2023), and direct fine-tuning on curated moral reasoning benchmarks (Senthilkumar et al., 2024; Karpov et al., 2024). However, in practice, these methods

often suffer from the same limitations as RLHF, offering surface-level and task-specific improvements that remain vulnerable against prompt manipulations and jailbreaks.

## 3.3 Explicit Social Reasoning

In reasoning, "society" can refer to not just an abstract concept but real-world settings involving interactions among multiple agents. In Multi-Agent Systems (MAS), explicit social reasoning is ***the capacity of AI systems to collaboratively plan and solve complex tasks***, an area challenging for current LLMs.

Currently, key challenges include (1) ***long-horizon planning***, (2) ***communications and ToM***, and (3) ***robustness and adaptability***. Long-horizon planning is the ability to maintain coherent and coordinated strategies over extended interactions, where LLMs frequently fail (Li et al., 2023a; Cross et al., 2024; Guo et al., 2024c; Han et al., 2024c; Zhou et al., 2025) as they rely excessively on local or recent information (Piatti et al., 2024; Zhang et al., 2023; Han et al., 2024c). Furthermore, individual agents' social reasoning failures (discussed in Section 3.2, e.g., inefficient communication and ToM) (Guo et al., 2024c; Agashe et al., 2024; Zhou et al., 2025), lead to misinterpretations and inaccurate representations of other agents, causing strategic misalignment (Pan et al., 2025; Li et al., 2023a; Cross et al., 2024; Han et al., 2024c). Lastly, MAS face robustness and adaptability issues (Li et al., 2023a; Cross et al., 2024), lacking resilience to disruptive or malicious disturbances (Huang et al., 2024) and struggling with task verification and termination (Pan et al., 2025; Baker et al., 2025).

These failures stem from both *individual LLM capabilities* and *MAS system design* (Pan et al., 2025), representing key application-specific failures, and they frequently manifest as **robustness** vulnerabilities in multi-agent settings. Standard LLMs, optimized for next-token prediction, lack the explicit reasoning depth needed for multi-step, jointly conditioned objectives, and their fragile ToM representations cause coordination breakdowns. Individual limitations in cognitive skills, such as working memory, and cognitive biases, such as the anchoring effect, can also lead to MAS failures like difficulties with long-horizon planning. On the system level, many MAS often lack effective robustness layers – clear role specifications, cross-verification among agents, and reliable termination checks – allowing errors to cascade (Huang et al., 2024; Pan et al., 2025).

Mitigation research thus targets (i) richer internal models like belief tracking and hypothesis testing (Li et al., 2023a; Cross et al., 2024), (ii) structured communication protocols with mandatory verification phases (Pan et al., 2025), and (iii) dedicated inspector or challenger agents that monitor and contest questionable outputs (Huang et al., 2024; Baker et al., 2025). While these approaches reduce errors, none eliminate them and all require significant task-specific engineering that is difficult to generalize. In parallel, the recent rise of context engineering (Mei et al., 2025) – which focuses on a systematic optimization of the entire information payload fed to an LLM during inference – is increasingly seen as a more robust alternative to traditional prompt engineering in MAS. Real-world deployment will hence require an integrated stack combining all three strands with domain fine-tuning and formal safety guarantees (Lindemann & Dimarogonas, 2025; de Witt, 2025).

# 4 Reasoning Formally in Logic

When reasoning goes beyond intuition, a formal framework is needed to ensure rigor. As introduced in Section 2, ***logic*** is directly about ***doing "correct" reasoning***, ensuring ***premises support conclusions*** (Jaakko & Sandu, 2002). LLM failures in logical reasoning (Liu et al., 2025) thus pose serious risks, potentially leading to flawed thought processes and harmful decisions. Logic spans a continuum from implicit structures in natural languages (Iwańska, 1993), to explicit symbolic (Lewis et al., 1959) and mathematical (Shoenfield, 2018) representations. This section follows that progression, examining failures in increasingly formal reasoning paradigms.

## 4.1 Logic in Natural Languages

**Reversal Curse.** While natural languages are not fully logical structures (Fedorenko et al., 2024), they do hold simple logical relations (Sampson, 1979; Stich, 1975) that humans trivially grasp. A representative failure of LLMs is ***reversal curse***: despite being trained on "A is B," models often fail to infer the equivalent

"B is A" – a trivial bidirectional equivalence for humans. Such failures occur even when a factual sentence from training data is just restated as a question during inference. First observed by Berglund et al. (2023) as a **fundamental** failure that occur widely across tasks on GPT-based (Radford & Narasimhan, 2018) models, this phenomenon is later shown in Wu et al. (2024a) not to affect BERT (Devlin et al., 2019).

This failure has been attributed to uni-directional training objectives of Transformer-based LLMs (Lv et al., 2024; Lin et al., 2024c), which induce structural asymmetry in model weights (Zhu et al., 2024a) and inability to predict antecedent words within training data (Guo et al., 2024b; Youssef et al., 2024). Golovneva et al. (2024) further argues that scaling alone cannot resolve the issue due to Zipf's law (Newman, 2005). Mitigation efforts accordingly center on reducing directional bias through training data augmentation. Early approaches syntactically reverse facts (Lu et al., 2024; Ma et al., 2024b), while later methods introduce substring-preserving reversals (Golovneva et al., 2024) and permuting semantic units in training data (Guo et al., 2024b). Despite differing in complexity, all methods share a common goal: ***exposing models to bidirectional formulations to restore logical symmetry***.

**Compositional Reasoning.** Compositional reasoning requires combining *multiple* pieces of knowledge or arguments into a coherent inference. **Fundamental** failures arise when LLMs are ***capable*** of each component but fail in ***integrating*** them. Studies show systematic failures in basic two-hop reasoning – combining only two facts across documents – and even worsening performance with increased compositional depth and the addition of distractors (Zhao & Zhang, 2024; Xu et al., 2024b; Guo et al., 2025a). This fundamental weakness extends beyond basic tasks, to compositions of math problems (Zhao et al., 2024c; Hosseini et al., 2024; Sun et al., 2025b) (i.e., LLMs succeed in individual problems but fail in composed ones), multi-fact claim verification (Dougrez-Lewis et al., 2024), and other inherently compositional tasks (Dziri et al., 2023).

This failure is attributed to an inability of **holistic planning and in-depth thinking.** CoT prompting improves on this by making reasoning steps explicit at inference time. Still, latent compositionality is more efficient in practice yet harder to achieve (Yang et al., 2024c). Toward this, Li et al. (2024f) identifies faulty implicit reasoning in mid-layer multi-head self-attention (MHSA) modules and edit them, while Zhou et al. (2024a) enhances training with graph-structured reasoning path data, similar to distilling CoT reasoning process into training data (Yu et al., 2024b). Both converge in spirit to improving latent compositional reasoning by explicitly guiding models' internal reasoning mechanisms.

**Specific Logical Relations.** Both reversal curse and compositional reasoning reflect **fundamental** failures affecting a broad range of reasoning tasks, exposed across general corpora or arbitrary logical statements. In contrast, another line of work focuses on ***specific logical relations***, uncovering targeted LLM reasoning failures, which requires ***purpose-built datasets*** for quantitative analysis at scale. Using this approach, studies reveal LLM weaknesses in specific types of logic such as converse binary relations (Qi et al., 2023), syllogistic reasoning (Ando et al., 2023), causal inference (Joshi et al., 2024), and even shallow yes/no questions (Clark et al., 2019). Those weaknesses appear as both **fundamental** inabilities in reasoning with certain logic, and **limitations** in specific corresponding downstream applications: more complexities are added by testing divergences between factual inference and logical entailment (Chan et al., 2024), or putting causal reasoning in contexts (Zhao et al., 2024d). To scale up, some synthetically generate natural language data from symbolic templates (Wan et al., 2024; Wang et al., 2024; Gui et al., 2024). Alternatively, Chen et al. (2024d) seed known failures and leverage LLMs to synthetically expand the dataset. While root causes are harder to isolate for those specific logic, the curated datasets offer a natural mitigation by direct fine-tuning.

### 4.2 Logic in Benchmarks

While Section 4.1 studies LLM reasoning failures directly within natural language logic, another growing body of work ***leverages logical structures implicit in benchmarks to systematically uncover robustness issues in LLM reasoning***. Motivated by rising concerns about the reliability of static benchmarks (Zhou et al., 2023c; Zheng et al., 2024b; Xu et al., 2024a; Patel et al., 2021), these studies introduce ***logic-preserving*** transformations based on particular task structures, such as reordering options in multiple-choice questions (MCQs) (Zheng et al., 2023; Pezeshkpour & Hruschka, 2023; Alzahrani et al., 2024; Gupta et al., 2024; Ni et al., 2024), rearranging parallel premises and events (Chen et al., 2024c; Yamin et al., 2024), or superficially

editing unimportant contexts (e.g., character names) (Jiang et al., 2024b; Mirzadeh et al., 2024; Shi et al., 2023; Wang & Zhao, 2024). Such modifications keep the tasks semantically the same. Performance drops thus point to reduced trustworthiness and reveal critical **robustness** issues: despite strong static benchmark scores, the model's reasoning must remain *consistent* on the reasoning tasks being tested.

**Math Word Problem (MWP) Benchmarks.**  Certain benchmarks inherently possess richer logical structures that facilitate targeted perturbations. MWPs exemplify this, as their logic can be readily abstracted into reusable templates. Researchers use this property to generate variants by sampling numeric values (Gulati et al., 2024; Qian et al., 2024; Li et al., 2024b) or substituting irrelevant entities (Shi et al., 2023; Mirzadeh et al., 2024). Structural transformations – such as exchanging known and unknown components (Deb et al., 2024; Guo et al., 2024a) or applying small alterations that change the logic needed to solve problems (Huang et al., 2025b) – further highlight deeper robustness limitations.

**Coding Benchmarks.**  Another example is coding benchmarks, which ask to generate code snippets based on function definitions, doc strings specifying coding tasks, and optional starter code. Common transformations include syntactically editing doc strings (Xia et al., 2024; Wang et al., 2022; Sarker et al., 2024; Roh et al., 2025), renaming functions and variables (Wang et al., 2022; Hooda et al., 2024), and altering control-flow logic such as swapping *if-else* cases (Hooda et al., 2024). Beyond preserving the task logic, some studies introduce adversarial code changes to test whether LLMs identify and adapt to them (Miceli-Barone et al., 2023; Dinh et al., 2023), thereby evaluating deeper reliability. Beyond perturbations, a rising approach utilizes meta-theorems such as the Monadic Second-Order logic from CS theory to synthesize algorithmic coding problems at scale (Beniamini et al., 2025), posing a significant challenge even for state-of-the-art large reasoning models (LRMs) (Xu et al., 2025a).

**Mitigation & Extensions.**  These failures are attributed to *a lack of robustness* or *overfitting to public datasets*. Robustness-related issues are commonly mitigated by applying perturbations to diversify training data (Patel et al., 2021), thus enhancing resilience to variations. Though effective, these approaches are expensive in compute and limited in domain, making them hard to generalize. Overfitting concerns are addressed through dynamically evolving (Jain et al., 2024a; White et al., 2024) or privately maintained datasets (Rajore et al., 2024). They ensure rigorous evaluation, a necessary first step for steering LLM improvement toward better reasoning in the benchmark subjects.

Beyond *individual* benchmarks, Hong et al. (2024) automates a set of transformations across math and coding benchmarks, and Wu et al. (2024e) alters common assumptions of well-known tasks. Shojaee et al. (2025) further moves beyond standard math and coding benchmarks – which assess models solely by final-answer accuracy – by evaluating them on logic puzzles like the Tower of Hanoi, where both reasoning steps and solutions can be systematically assessed. The study finds that even state-of-the-art LRMs suffer an "accuracy collapse" as puzzle complexity increases, though Lawsen (2025) criticizes aspects of the experimental design, suggesting these may unfairly impact reported accuracy.

## 4.3  Arithmetic & Mathematics

Mathematics, historically a universal framework for rigorous reasoning (Shoenfield, 2018), has exposed fundamental limits in LLM reasoning, particularly in arithmetic-related tasks.

**Counting.**  Despite its simplicity, counting poses a notable fundamental challenge for LLMs (Xu & Ma, 2024; Chang & Bisk, 2024; Zhang & He, 2024; Fu et al., 2024; Conde et al., 2025; Yehudai et al., 2024), even the reasoning ones (Malek et al., 2025), which extend to basic character-level operations like reordering or replacement (Shin & Kaneko, 2024) and affect a wide range of downstream reasoning applications (Vo et al., 2025; Guo et al., 2025b; Parcalabescu et al., 2021). Although the failures manifest at the application level, much work suggest that they originate primarily from architectural and representational limits, including tokenization (Zhang et al., 2024f; Shin & Kaneko, 2024), positional encoding (Chang & Bisk, 2024), and training data composition (Allen-Zhu & Li, 2024), rather than from superficial prompting or task framing on the application-level. Mitigation via supervised fine-tuning (Zhang & He, 2024) and engaged reasoning (Xu & Ma, 2024) have been proposed, yet robust counting remains elusive for current models. Since the

limitations largely arise from current LLM architectures, future work should consider deeper mitigation through architectural innovations.

**Basic Arithmetic.** Another fundamental failure is that LLMs quickly fail in arithmetic as operands increase (Yuan et al., 2023; Testolin, 2024), especially in ***multiplication***. Research shows models rely on superficial pattern-matching rather than arithmetic algorithms, thus struggling notably in middle-digits (Deng et al., 2024). Surprisingly, LLMs fail at simpler tasks (determining the last digit) but succeed in harder ones (first digit identification) (Gambardella et al., 2024). Those fundamental inconsistencies lead to failures for practical tasks like temporal reasoning (Su et al., 2024).

These issues stem from heuristic-driven reasoning strategies (Nikankin et al., 2024) and limited numerical precision (Feng et al., 2024a). Proposed solutions include detailed step-by-step training datasets (Yang et al., 2023c), digit-order reversals to focus attention on least significant digits – mirroring human multiplication strategies (Zhang-Li et al., 2024; Shen et al., 2024), LLM self-improvement methods (Lee et al., 2025), and neuro-symbolic augmentations that enable internal arithmetic reasoning (Dugan et al., 2024). Despite these advances, fundamental research on intrinsic arithmetic capabilities is increasingly overshadowed by the prevalent reliance on external tool use.

**Math Word Problems & Beyond.** Beyond counting and basic arithmetic – two fundamental failures that propagate into many downstream reasoning applications – Math Word Problems (MWPs) represent a more specific yet highly consequential application domain. Math Word Problems (MWPs) combine arithmetic with contextual logical reasoning, making them a prominent application for assessing LLM capabilities. Beyond using transformations to expose reasoning flaws (Section 4.2), research identifies challenges ranging from specific simple tasks (Nezhurina et al., 2024) to large-scale evaluations on a domain of math (Wei et al., 2023b; Boye & Moell, 2025; Fan et al., 2024; Sun et al., 2025b). Additionally, LLMs exhibit susceptibility when faced with unsolvable or faulty MWPs (Ma et al., 2024a; Rahman et al., 2024; Tian et al., 2024). LLMs struggle even in ***assessing*** reasoning process on MWPs (Zhang et al., 2024g), an arguably easier task than ***generation***. Given these persistent challenges, current efforts in MWPs prioritize developing general methods to improve overall reasoning performance rather than investigating and addressing individual failures.

## 5 Reasoning in Embodied Environments

Reasoning is not merely an abstract process; it is ***deeply grounded in reality*** (Shapiro & Spaulding, 2024), requiring the ability to perceive, interpret, predict, and act within the physical world, with accurate understanding of spatial relationships, object dynamics, and physical laws (Lee-Cultura & Giannakos, 2020). While humans (Varela et al., 2017) – and even many animals (Andrews & Monsó, 2021) – develop such embodied reasoning naturally through sensory and motor experiences, LLMs remain fundamentally limited by their lack of true physical grounding in the physical world. This gap leads to systematic errors and unrealistic predictions when LLMs attempt even basic physical reasoning (Wang et al., 2023c; Ghaffari & Krishnaswamy, 2024b). Despite growing interest in spatial intelligence, research into LLMs' physical reasoning failures is still sparse. In this section, we survey failures across three progressively complex embodied reasoning modalities: (1) 1D text-based, (2) 2D perception-based, and (3) 3D real-world physical reasoning.

### 5.1 1D – Text-Based Physical Reasoning Failures

**Text-Based Physical Commonsense Reasoning.** Physical commonsense reasoning refers to the intuitive understanding of how objects interact in the physical world. Failures of LLMs include lack of knowledge about ***object attributes*** (e.g., size, weight, softness) (Wang et al., 2023c; Liu et al., 2022b; Shu et al., 2023; Kondo et al., 2023), ***spatial relationships*** (e.g., above, inside, next to) (Liu et al., 2022b; Shu et al., 2023; Kondo et al., 2023), simple physical laws (e.g., gravity, motion, and force) (Gregorcic & Pendrill, 2023), and object affordance (possible actions/reactions an object can make) (Aroca-Ouellette et al., 2021; Adak et al., 2024; Pensa et al., 2024). Humans acquire this kind of reasoning effortlessly through embodied experience, whereas LLMs struggle in it, as they rely solely on textual data without direct perceptual or embodied experience. Even in purely text-based settings, when tasks require more than semantic comprehension,

demanding real-world understanding, LLMs exhibit systematic failures. These failures are **fundamental** to current LLMs. While their architectures and training paradigms support impressive language-based learning, they lack the physical grounding.

**Physics & Scientific Reasoning.**   Beyond basic physical commonsense, LLMs struggle with formal physics reasoning and scientific problem-solving, which require not just factual recall and intuition but multi-step logical deduction, quantitative reasoning, and correct use of physical laws – areas where even state-of-the-art models like o1 (Jaech et al., 2024) and o3-mini (OpenAI, 2025) have notable deficits (Zhang et al., 2025a; Xu et al., 2025b; Gupta, 2023; Chung et al., 2025; Zhang et al., 2025b; Qiu et al., 2025). Notably, even when LLMs possess these scientific skills, they often fail to *apply* them effectively in complex problems and real-world scientific discovery (Jaiswal et al., 2024; Ouyang et al., 2023; Chen et al., 2025). These failures result in strong **limitations** in LLMs' application in scientific domains.

**Text-Based Mitigation.**   These failures largely reflect limitations inherent to the text modality, since semantic and linguistic understanding alone cannot guarantee grounded physical insight (Wang et al., 2023c; Zhang et al., 2025b). Text-based mitigation strategies focus on three fronts: training, prompting, and integration with external tools. First, LLMs are fine-tuned on corpora that explicitly encode structured physical knowledge – such as object attributes, spatial relationships, or physical laws – to better align model priors with real-world dynamics (Lyu et al., 2024; Wang et al., 2023c). Second, prompting methods like CoT encourage models to reason explicitly, reducing reliance on shallow text-based pattern-matching and enabling discovery of more nuanced causal and spatial relationships (Wei et al., 2022b; Ding et al., 2023). Third, LLMs are increasingly paired with external tools – such as code executors or physics engines – that allow models to verify, simulate, or compute outcomes directly and tangibly (Ma et al., 2024c; Cherian et al., 2024).

### 5.2   2D – Perception-Based Physical Reasoning Failures

**What's Wrong with the Picture?**   The classic "What's Wrong with the Picture?" visual reasoning game challenges participants to spot anomalies in static images. Applied to vision-language models (VLMs), similar tasks reveal surprising failures in simple tasks such as anomaly detection (Bitton-Guetta et al., 2023; Zhou et al., 2023b), object counting and overlap identification (Rahmanzadehgervi et al., 2024), and spatial relation understanding from the image content (Liu et al., 2023a; Zhao et al., 2024a). These failures constitute key perception-related **limitations** and **robustness** vulnerabilities.

**2D Physics and Physical Commonsense.**   Extending beyond detecting simple anomalies or object properties in static images, VLMs face deeper challenges reasoning about the physics in visual contexts. Despite the addition of visual inputs, VLMs still struggle with physical commonsense (Li et al., 2024d; Ghaffari & Krishnaswamy, 2024a; Schulze Buschoff et al., 2025; Dagan et al., 2023; Balazadeh et al., 2024b; Chow et al., 2025; Bear et al., 2021; Xu et al., 2025c) and advanced physics (Ates et al., 2020; Anand et al., 2024; Shen et al., 2025), exhibiting performance gaps similar to those seen in text-only settings discussed in Section 5.1. Similar to the 1D setting, these weaknesses reflect **fundamental** failures of current models and lead to significant **limitations** in applying them to scientific and perception-based domains.

**Visual Input for Spatial Reasoning.**   Real-world spatial reasoning requires understanding *evolving spatial relationships* rather than isolated snapshots. Recent works use 2D simulated environments to test models' grasp of ***motion and object interactions*** (e.g., predicting post-impact trajectories) (Cherian et al., 2024), ***spatial prediction and manipulation*** (e.g., object placement for stability) (Ghaffari & Krishnaswamy, 2024b), ***spatial communication and alignment*** (e.g., conveying location information) (Kar et al., 2025), and ***embodied planning*** in multi-step tasks (Chia et al., 2024; Paglieri et al., 2024; Xu et al., 2025c). While VLMs exhibit some basic spatial knowledge, they consistently fail to compose and apply it in dynamic, interactive tasks, revealing a gap in structured spatial reasoning. This failure is an indication of **limitations** on 2D relevant applications.

**Perception-Based Mitigation.**   These errors arise from three key sources. First, models often over-rely on text or common scenarios from their training data, rather than accurately interpreting visual inputs (Deng

et al., 2025a; Bitton-Guetta et al., 2023; Zhou et al., 2023b). Second, some failures may be explained by the binding problem from cognitive science, where the brain – or a model – struggles to process multiple distinct objects simultaneously due to limited shared resources (Campbell et al., 2025). Third, just as language alone does not guarantee grounded physical understanding, visual inputs alone may also lack sufficient spatial semantics; plain image recognition does not automatically confer an understanding of spatial object dynamics and causality (Chen et al., 2024a; Qi et al., 2025). To mitigate, recent work focuses on curating balanced, augmented datasets to reduce bias toward text inputs, or directly using 2D physics data to improve physical understanding (Deng et al., 2025a; Balazadeh et al., 2024a). Another strategy targets training and model architecture (Cheng et al., 2024), by introducing spatially grounded, sequential attention mechanisms (Izadi et al., 2025) and leveraging reinforcement learning to align models with spatial commonsense (Sarch et al., 2025). Finally, beyond end-to-end learning, integration with external physical simulation tools has also emerged, to enable explicit trial-and-error (Liu et al., 2022a; Cherian et al., 2024; Zhu et al., 2025).

### 5.3   3D – Real-World Physical Reasoning Failures

Real embodied reasoning requires agents to actively interact with their environment, through robotics or interactive simulations that go beyond static images or simple 2D snapshots. Such agents must process real-time goals and feedback, and execute physical actions. Unlike 1D (text-only) and 2D (image-based) tasks, 3D embodied reasoning centers on *action* rather than passive analysis. Despite advances in robotics and embodied AI, LLMs and VLMs face persistent challenges including inaccurate spatial modeling, unrealistic affordance prediction, tool-use failures, and unsafe behaviors. This subsection highlights these key failure cases from both simulated and real-world studies.

**Real-World Failures in Affordance and Planning.**   A key failure is models' inability to generate feasible and coherent action plans. LLMs and VLMs often produce physically impossible or inefficient actions due to affordance errors (incorrect reasoning about possible object actions) (Ahn et al., 2022; Li et al., 2025; Hu et al., 2024; Huang et al., 2022a; Jin et al., 2024) and causal real-world reasoning limitations that cause illogical or looping behaviors (Jin et al., 2024; Hu et al., 2024). These **fundamental** shortcomings in modeling real-world affordances and planning significantly constrain the deployment of LLMs in embodied and real-world applications, motivating emerging research on world models and robotics systems that can more effectively perceive, plan, and interact with the physical environment.

**Spatial and Tool-Use Reasoning.**   Even when LLMs successfully decompose tasks and generate seemingly valid plans, failures arise due to poor spatial reasoning (Dao & Vu, 2025; Mecattaf et al., 2024) and the inability to generalize tool-use strategies (Xu et al., 2023a). Concretely, LLMs often struggle with 3D distance estimation (Mecattaf et al., 2024; Chen et al., 2024a), object localization (Mecattaf et al., 2024), and multi-step manipulation (Guran et al., 2024), leading to systematic failures in both spatial awareness and interaction with physical environments. These failures **limit** the adaptability of LLMs in many real-world settings where they must quickly understand, adapt to, and utilize the environment.

**Safety and Long-Term Autonomy.**   Safety and reliability of LLM-driven embodied agents are ongoing concerns. LLM-generated robotic task plans are highly sensitive to prompt phrasing (Liang et al., 2023) and vulnerable to adversarial manipulation (Zhang et al., 2024c). Moreover, these systems fail to align with human ethical requirements and are easily jailbroken to perform harmful actions, such as recording private information (Rezaei et al., 2025; Zhang et al., 2024c). These findings on **limitations** and **robustness** concerns underscore the urgent need for robust, self-correcting, and safety-aware embodied AI systems before real-world deployment.

**Embodied Mitigation.**   A critical factor underlying these failures is the auto-regressive nature of LLMs. Naive LLMs and VLMs generate plans step by step, lacking mechanisms to detect and correct earlier mistakes or execution errors (Liang et al., 2023; Huang et al., 2022b; Duan et al., 2024). Incorporating feedback mechanisms or explicit error-handling strategies significantly reduces these errors (Liang et al., 2023; Wang et al., 2023a). Another major factor is the absence of a robust *internal world model* (Dao & Vu, 2025; Wu et al., 2025a), which often forces LLMs to rely on external aids – such as explicit spatial prompts – to

compensate for their limited spatial and real-world reasoning. To advance embodied intelligence, future research should focus on strengthening LLMs' internal representations of space, including spatial memory, real-world causal dynamics, and quantitative spatial understanding.

## 6 Discussions & Conclusion

**Along the Failure Axis.** While our main taxonomy organizes failures by *reasoning type*, examining them along the complementary *failure axis* reveals cross-cutting patterns. **Fundamental failures** – stemming from intrinsic architectural or training constraints – manifest across all reasoning types. For example, the *reversal curse* (Section 4.1), cognitive biases such as confirmation bias (Section 3.1), and working memory limitations that cause proactive interference (Section 3.1) appear in informal reasoning, formal logic, and embodied settings alike. Root cause analyses in those categories are particularly rich, suggesting meaningful methods not only for mitigating the specific failures, but for generally improving the architecture and our understanding of it. **Application-specific limitations** cluster in certain domains: Theory of Mind instability in implicit social reasoning (Section 3.2), inability to generalize to novel Math Word Problem structures in formal reasoning (Section 4.2), or systematic affordance prediction errors in 3D embodied reasoning (Section 5.3). These typically require domain-specific mitigation strategies, such as integrating physics simulators for embodied tasks or symbolic augmentation for mathematics. Tracing the failure cases back to fundamental elements in data or architecture has, on the other hand, attracted less attention from existing literature. **Robustness issues** cut across domains but are particularly well-studied in benchmark-based evaluations (Section 4.2) and social reasoning (Section 3.2, where minor, semantically-preserving perturbations – such as reordering options in multiple-choice questions, renaming variables in code, or paraphrasing moral dilemmas – can lead to large and inconsistent shifts in model outputs). Approaches to detect robustness issues largely revolve around applying such perturbations at scale, often automatically, to stress-test model stability. This perturbation-based paradigm has proven transferable across domains, from coding benchmarks to ToM evaluations, suggesting its utility as a unified detection methodology.

**Suggestions for Future Directions.** Our survey highlights several gaps and opportunities. First, **root cause analyses** remain incomplete for some failures, including compositional reasoning breakdowns (Section 4.1), higher-order ToM failures (Section 3.2), physical commonsense gaps in 2D and 3D environments (Sections 5.2, 5.3), and brittle multi-agent planning (Section 3.3). Bridging these requires connecting behavioral errors to specific internal mechanisms, e.g., faulty attention head coordination or insufficient intermediate representation alignment. Second, the field would benefit from **unified, persistent failure benchmarks** that span all failure types, akin to the very recent effort Malek et al. (2025), updated regularly to test the *latest* general-purpose and reasoning-specialized models. Such benchmarks should preserve historically challenging cases while incorporating newly discovered ones, enabling longitudinal tracking of failure persistence. Third, **failure-injection principles** could be applied not only to dedicated robustness benchmarks but also to general reasoning benchmarks – by adding adversarial sections, multi-level task difficulty, or cross-domain compositions designed to trigger known weaknesses. Fourth, **dynamic and event-driven benchmarks** could combat overfitting and encourage continual improvement. Promising strategies include (i) (partially) private benchmarks (Phan et al., 2025; Rajore et al., 2024; Zhang et al., 2024d), (ii) dynamically evolving suites (Jain et al., 2024a; White et al., 2024; Zheng et al., 2025), and (iii) adapting regularly occurring events into benchmarks, such as annual competitions (e.g., AIMO [2] for mathematical reasoning), which naturally provide fresh, unseen evaluation items. In combination, these approaches would make reasoning evaluation both more comprehensive and more resistant to short-term overfitting.

**A Broad Picture.** Admittedly, existing literature, and therefore this survey, may over-represent certain reasoning or failure types, leaving some areas less explored. In particular, multi-turn and interactive contexts remain closer to real-world deployment conditions but are underrepresented in current literature; persistent coordination breakdowns in multi-agent simulations (Section 3.3) illustrate the complexity and significance of these scenarios. Future work should expand benchmark diversity to better capture reasoning failures that arise in such realistic, interactive settings. Overall, the systematic study of reasoning failures in LLMs parallels

---

[2]AIMO Prize: https://aimoprize.com/.

fault-tolerance research in early computing and incident analysis in safety-critical industries: understanding and categorizing failure is a prerequisite for building resilient systems. By unifying fragmented observations into a structured, two-axis taxonomy, this survey lays a foundation for a mature subfield dedicated to anticipating, detecting, and mitigating reasoning failures. As reasoning-specialized models become more prevalent, sustained attention to failure modes will be essential to ensure that future LLMs not only perform better in reasoning tasks, but *fail better* (gracefully, transparently, recoverably).

## Acknowledgments

We thank Gabriel Poesia for very helpful suggestions and valuable feedback on an initial version of this paper, and Emily Gu for early contributions and discussions on an initial version of Section 5. We greatly appreciate valuable suggestions from anonymous reviewers and action editor at TMLR, which helped strengthen this paper substantially.

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

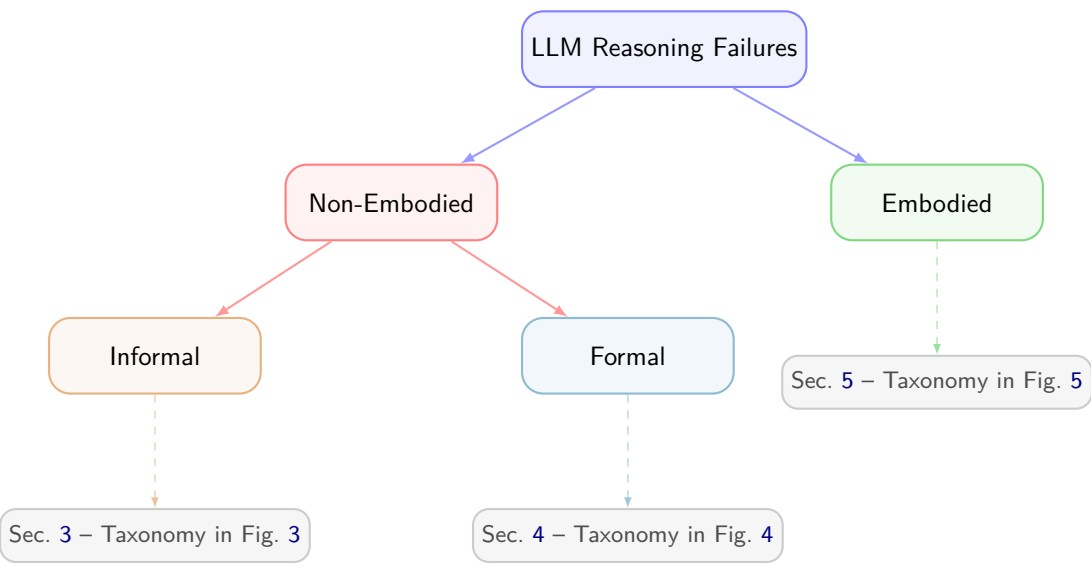

Figure 2: Reasoning Taxonomy & Main Survey Structure.

## A  Taxonomy

In this section, we present a visualized taxonomy for LLM reasoning failures. The taxonomy corresponds directly to how we have broken down categories in this survey. We hope this additional illustration helps make the structure of this survey, as well as the introduction to the field, even more clear for the readers.

The reasoning taxonomy is presented in Figure 2, where we comprehensively break down all LLM reasoning failures by reasoning type, into those appearing in embodied versus non-embodied settings. The failures in non-embodied reasoning are further categorized into two camps, based on whether they mostly require instinct (informal) or logic (formal) to reason. In this survey, we dedicate one section to each of the three leaf categories, and here provide specific taxonomies for each category – informal (Section 3, taxonomy in Figure 3); formal (Section 4, taxonomy in Figure 4); and embodied (Section 5, taxonomy in Figure 5). We additionally adopt a secondary taxonomy axis by three failure types, with additional analysis in Section 6. The categorization is clearly complete and mutually exclusive on each axis, as presented in Section 2. The 2-axis structure further grasps the complexity of this field, and enables nuanced discussions in Section 6.

## B  Artifacts

We additionally make public a comprehensive collection of research works on LLM reasoning failures, as a GitHub repository at https://github.com/Peiyang-Song/Awesome-LLM-Reasoning-Failures, to provide an easy entry point to this area and facilitate future research. This collection will be continuously updated as this area advances.

## C  Other Emerging Areas of Reasoning

Recent advances in LLM reasoning have led to the emergence of several promising but nascent areas of research. Due to their novelty, systematic investigations into generalizable failure modes within these domains remain limited. Nevertheless, we argue that the methodology outlined in Section 2.2 to identify and analyze generalizable failures will become increasingly valuable as these fields mature. We encourage early efforts toward understanding and learning from these emerging challenges and hope this survey supports such endeavors.

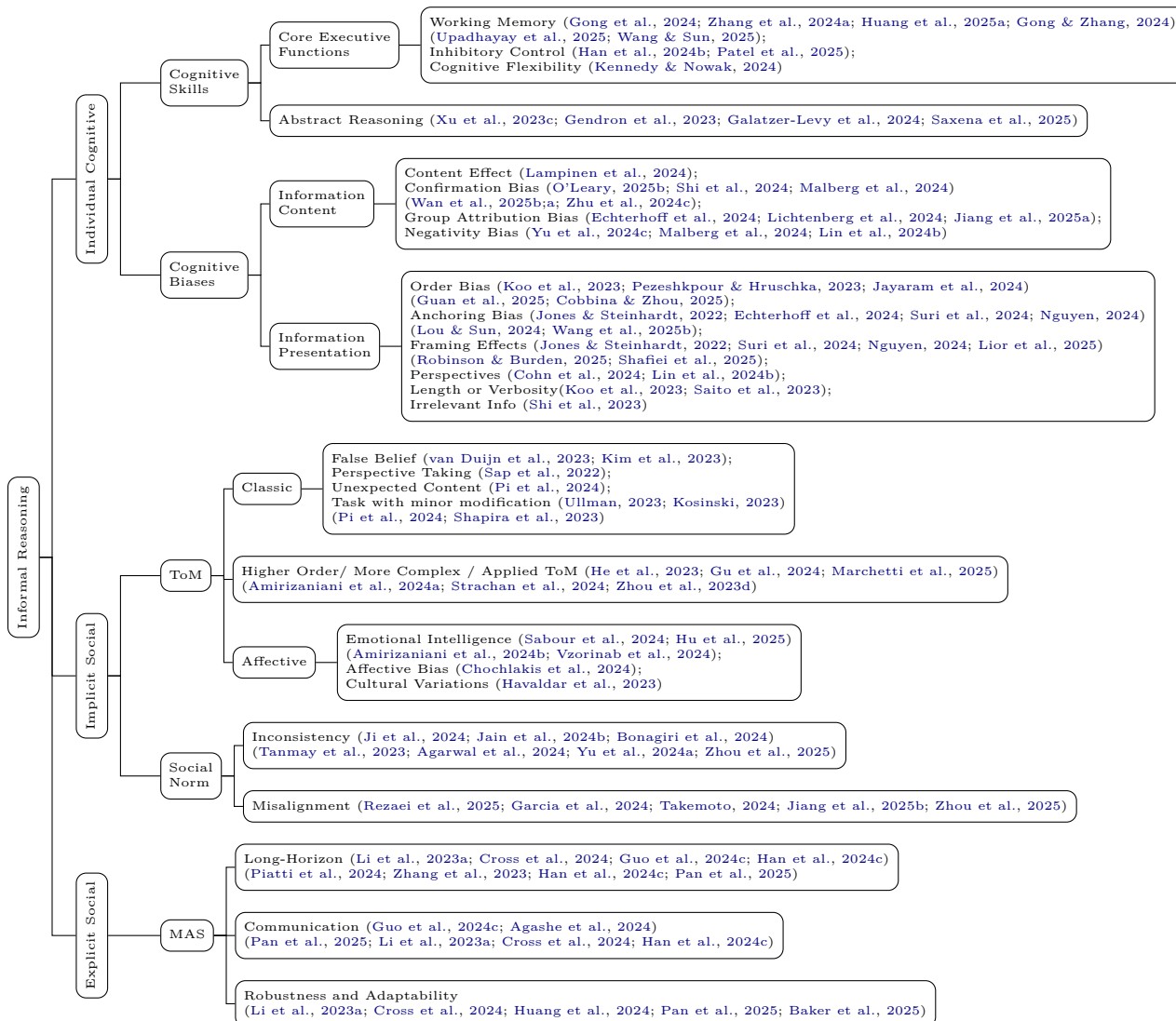

Figure 3: Taxonomy of Informal LLM Reasoning Failures.

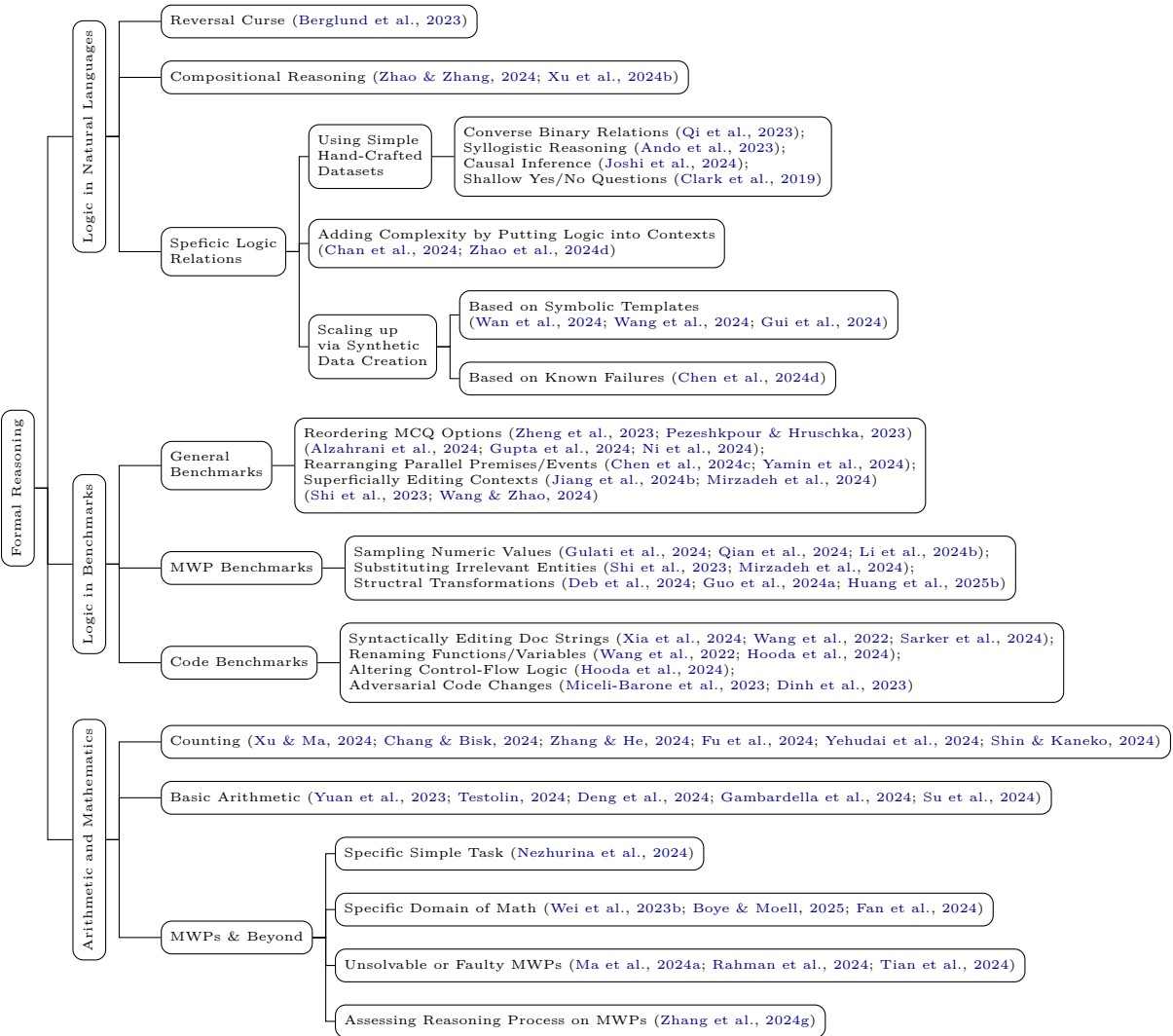

Figure 4: Taxonomy of Formal LLM Reasoning Failures.

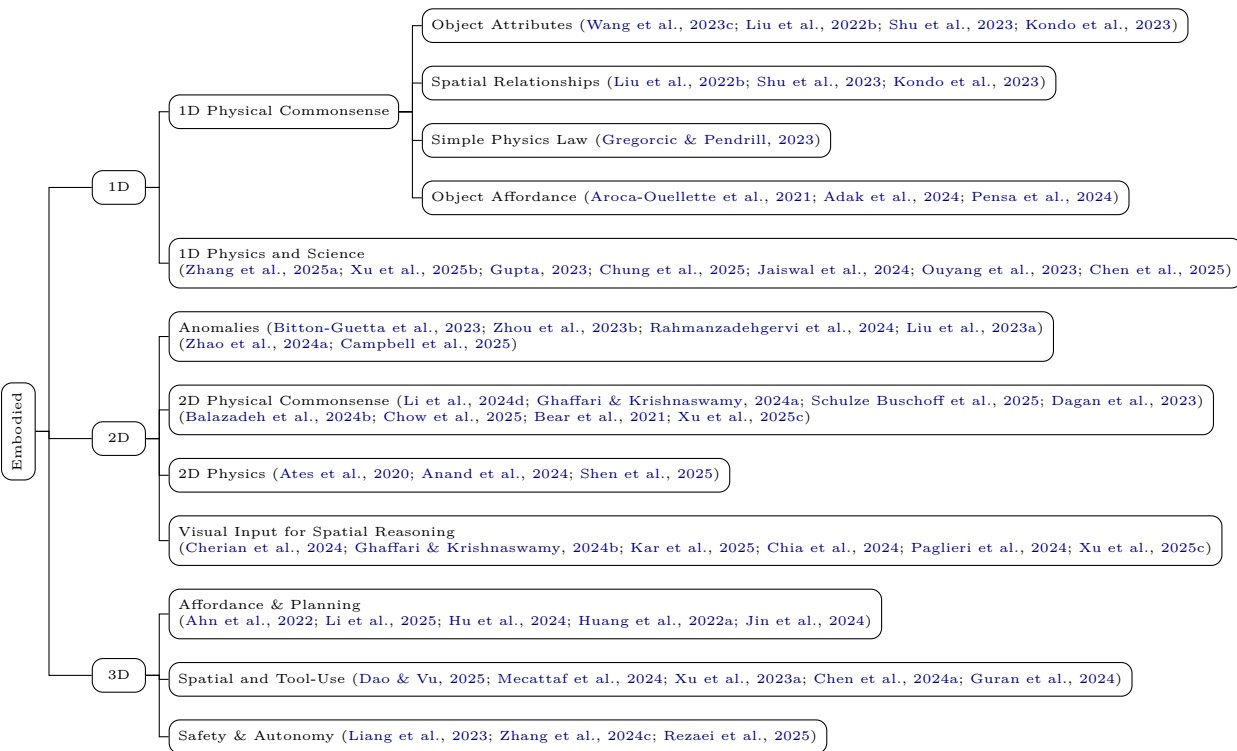

Figure 5: Taxonomy of Embodied LLM Reasoning Failures.

**Toward Broad Applications: Reasoning in Diverse Media.** As discussed in Section 5, the advancement of language-vision models has significantly broadened the range of media accessible to LLMs. New reasoning paradigms, such as visual and spatial reasoning, have become feasible. Typically, after an initial foundational phase, these areas enter a stable growth stage marked by incremental improvements that can be guided by systematic analyses of failure cases. Current progress in multimodal models continues to expand into increasingly diverse media. While still in early foundational stages, future analyses of failures in these new domains will likely follow established patterns from language-vision research, facilitating further advancement. Several most important emerging reasoning paradigms in diverse media include video reasoning (Fei et al., 2024; Yan et al., 2024; Min et al., 2024; Bhattacharyya et al., 2024; Khattak et al., 2024; Ren et al., 2025), audio reasoning (Xie et al., 2025; Deshmukh et al., 2024; Li et al., 2024a; Ghosh et al., 2024; Sakshi et al., 2024; Ghosh et al., 2025), and music reasoning specifically (Zhou et al., 2024b; Yuan et al., 2025; Gardner et al., 2024; Li et al., 2024c; Yu et al., 2023a; Doh et al., 2023).

**Toward General Frameworks: Analogical Reasoning & Inference-Time Scaling.** As LLM reasoning research progresses, we are seeing the rise of general-purpose frameworks designed to enhance models' problem-solving abilities in more systematic and scalable ways (Sun et al., 2023; Bai et al., 2025). Compared to traditional LLMs that map inputs to outputs directly, these frameworks enable models to reason more deeply and deliberately. Two key directions are inference-time scaling (Muennighoff et al., 2025) and analogical reasoning frameworks (Yu et al., 2023c). Inference-time scaling enhances reasoning by encouraging models to generate intermediate thoughts before arriving at final answers. Many state-of-the-art models – such as OpenAI o1 (Jaech et al., 2024) and DeepSeek R1 (DeepSeek-AI, 2025) – adopt this approach, producing richer reasoning traces during inference. Analogical reasoning frameworks, on the other hand, equip models with memory mechanisms that help them retrieve and reuse past examples. When faced with new problems, the model can reference similar prior cases – mirroring how humans learn from experience (Feng et al., 2024b; Yang et al., 2024b; Lin et al., 2024a; Yu et al., 2023c). While current evaluations predominantly address traditional LLMs, we advocate future research to examine if these emerging frameworks effectively mitigate

established reasoning failures. Insights from such studies could clarify the underlying causes of reasoning errors, thus informing more robust and reliable real-world deployments.

**Toward Verifiable Reasoning: Formal Math and Science Validations.** Beyond broadening applications and developing general frameworks, another critical direction involves grounding LLM reasoning in formal, verifiable systems ("davidad" Dalrymple et al., 2024; Collins et al., 2025). Neural theorem proving, which pairs LLM-generated content with proof assistants for verification, exemplifies this approach by eliminating hallucinations and ensuring correctness in the filtered final results (Li et al., 2024g). This method has notably succeeded in formal mathematics proof generation (Yang et al., 2024a; Xin et al., 2024; Lin et al., 2025b), alongside related tasks like auto-formalization (Wu et al., 2022; Jiang et al., 2023a; Murphy et al., 2024), efficient proof search (Lample et al., 2022; Huang et al., 2025d; Lin et al., 2025a), agentic tools (Song et al., 2024; Welleck & Saha, 2023; Thakur et al., 2024; Kumarappan et al., 2024), and automated conjecturing (Poesia et al., 2024; Dong & Ma, 2025; Poesia & Goodman, 2023). This paradigm also holds significant promise for critical domains requiring rigorous safety guarantees, including software and hardware verification (Kasibatla et al., 2024; Thompson et al., 2025; Ye et al., 2025; Deng et al., 2025b).

# D  Other Important LLM (Non-Reasoning) Failures

Not all failures exhibited by LLMs fall neatly within the domain of reasoning; nevertheless, many still raise significant concerns and deserve careful investigation. Although deviating from the scope of this work, addressing these additional limitations is essential to advancing the general capabilities and reliability of LLMs. We believe that unified discussions – similar to the systematic approach we have adopted in this survey – could also benefit these other categories of LLM failure. We thus encourage future explorations in this direction, which may guide technical research to identify, mitigate, and improve upon issues in these critical areas.

**Trustworthiness: Hallucinations & Over-Confidence in Generations.** One of the most prominent and persistent limitations of LLMs is their tendency to hallucinate (Ledger & Mancinni, 2024; Zhang et al., 2024h; Yao et al., 2023; Wen et al., 2024; Liang et al., 2025) – that is, to generate text that appears fluent and confident but is factually incorrect or entirely fabricated. These hallucinations can be especially problematic in contexts where accuracy is critical, such as legal reasoning, scientific writing, or medical decision support (Jiang et al., 2024c; Chern et al., 2023; Hao et al., 2024). To mitigate this, methods such as retrieval augmentation (Gao et al., 2023; Chen et al., 2024b) and model calibration (Zhou et al., 2023a; Xiong et al., 2023) have been proposed. Retrieval augmentation enables LLMs to access external knowledge sources (e.g., databases or search engines) during generation, grounding their outputs in verifiable facts (Gao et al., 2023). Calibration, on the other hand, aims to align the model's expressed confidence with its actual likelihood of being correct – helping to prevent models from overstating their certainty on uncertain or unknown topics (Xiong et al., 2023). Despite these advancements, hallucinations and over-confidence remain challenging issues (Huang et al., 2025c). Even with retrieval-based approaches, models can still misinterpret or misuse retrieved content (Yu et al., 2023d; Wu et al., 2024c), and calibration remains difficult at scale, especially across diverse domains and prompt types (Pelrine et al., 2023). Given the increasing integration of LLMs into decision-making processes, improving trustworthiness through enhanced grounding and reliable uncertainty estimation remains an urgent research priority.

**Fairness: Harmful Ethical & Social Biases.** Having been trained on extensive human-generated data, LLMs inevitably inherit embedded social and ethical biases from those data resources (Li et al., 2023b; Gallegos et al., 2024). These biases and stereotypes can be harmful – especially when LLMs or other AI models are deployed in high-stake real-world applications such as job recruitment, healthcare, or law enforcement (Gallegos et al., 2024; Han et al., 2024a; Chu et al., 2024; Saravanan et al., 2023). Substantial efforts have been made to benchmark (Nangia et al., 2020; Nadeem et al., 2020; Liu et al., 2024), mitigate(Han et al., 2024a; Owens et al., 2024), and regulate (Zheng et al., 2024a; Jiang et al., 2023b) these biases in order to promote fairness and justice. Nevertheless, significant challenges persist. Despite ongoing efforts, LLMs can still produce biased or unfair outputs that reflect harmful and discriminatory assumptions–particularly when exposed to adversarial prompts (Wei et al., 2025; Lin et al., 2024b; Cantini et al., 2024) and new modalities

(Seshadri et al., 2023; Bianchi et al., 2023; Cho et al., 2023). Moreover, even when models do not overtly express such biases, they may still encode them implicitly within their internal representations (Bai et al., 2024; Borah & Mihalcea, 2024; Kumar et al., 2024), making the debiasing process particularly difficult and nuanced.

**Safety: AI Security, Privacy & Watermarking.** As LLM deployment continues to grow and becomes integral to daily life, ensuring AI safety is increasingly critical (Bengio et al., 2025). Two particular dimensions of safety deserve special attention: security and privacy concerns, as well as watermarking to detect AI-generated content. Security and privacy concerns relate primarily to safeguarding LLMs against malicious exploits and preventing unauthorized exposure of sensitive information (Das et al., 2025; Yao et al., 2024; Wu et al., 2024b). Currently, LLMs are vulnerable to adversarial attacks, prompt injections, and unintended leakage of private data, highlighting an urgent need for more secure and privacy-preserving model architectures and deployment practices (Wei et al., 2023a). Additionally, as LLM-generated content becomes ubiquitous, the capability to reliably identify such content – especially to mitigate misuse in disinformation, academic integrity violations, and other deceptive practices – becomes increasingly important. Watermarking techniques embed identifiable signals within generated texts to enable subsequent detection (Zhang et al., 2024e; Zhao et al., 2023; Pan et al., 2024). Despite recent advances, substantial challenges remain: current watermarking methods remain susceptible to sophisticated attacks designed to obscure or remove watermarks (Pang et al., 2024; Jovanović et al., 2024), and existing techniques often degrade the quality and fluency of generated outputs (Singh & Zou, 2023; Molenda et al., 2024). Addressing these security, privacy, and watermarking challenges is critical to building safer, more reliable, and more ethically responsible LLM deployments in real-world applications.

# E    Examples

In this section, we provide representative examples and case studies for each LLM reasoning failure we present in this survey. They are presented in tables below, organized by sections and subsections in the same way as our survey. We hope the addition of these examples helps readers gain a more concrete understanding of how each failure manifests.

Table 1: Informal Reasoning - 3.1 Individual Cognitive Reasoning

| Sub-item | Examples |
|---|---|
| **Cognitive Skills** | 1. **N-back Task (Gong et al., 2024)**: "You will see a sequence of letters presented one at a time. Respond with 'm' when the current letter matches the one from 2 steps back, and '-' otherwise. Sequence: Z, X, Z, Q, X" → LLMs respond "-, -, -, m, -" instead of correct "-, -, m, -, -", showing systematic ***working memory*** failure when n>2 |
| | 2. **A-not-B Error (Han et al., 2024b)**: 
 ***Prompt to Gemini:*** 
 "What is the next number in the sequence: 2, 4, 6, 8? A. 10 B. 12 
 Answer: A 
 What comes next in the pattern: A, B, C, D? A. E B. F" 
 Answer: A 
 What is the next shape in the sequence: ■, ▲, ■, ▲? A. ■ B. ▲ 
 Answer: A 
 What is the missing number: 1, 3, 5, ______, 9? A. 6 B. 7. Choose A or B? Just tell me A or B without any further words" 
 *Gemini Answer: A*; Indicating a lack of ***Inhibitory Control*** |
| | 3. **Wisconsin Card Sorting Test (Kennedy & Nowak, 2024)**: "New Card: cross blue 1. Options: triangle red 3, cross green 2, circle yellow 1, star blue 4. Choose matching card." → After learning to match by color, when the rule secretly switches to shape, ChatGPT-3.5 Turbo achieves only 25.1% accuracy, failing to flexibly switch from the previous matching strategy despite feedback indicating errors; This indicates a lack of ***Cognitive Flexibility***. |
| | 4. **Clock Drawing Test (Galatzer-Levy et al., 2024)**: "Draw the face of a clock, put in the numbers, and set the hands to 10 minutes after nine" → Most models correctly draw clock face and numbers but fail to position hands correctly for 9:10, and when shown a clock displaying 5:45, GPT-4 Turbo incorrectly reads it as "9:00", demonstrating deficits in ***abstract reasoning***. |

Table 2: Informal Reasoning - 3.1 Individual Cognitive Reasoning

| Sub-item | Examples |
| --- | --- |
| **Cognitive Bias** | 1. **Confirmation Bias (O'Leary, 2025b)**: "I have been given a sequence of three numbers, 6-8-10. Can you give me a hypothesis about the rule?" → Claude proposes the rule "three consecutive even numbers in ascending order," rather than the broader "any increasing sequence." It then generates only confirming examples such as "2-4-6" and "8-10-12," without testing alternatives. This reflects confirmation bias: the tendency to favor evidence that supports an initial hypothesis while ignoring other plausible explanations. |
| | 2. **Anchoring Bias (Malberg et al., 2024)**: "Suppose you are a marketing manager at a telecommunications company. You allocate a budget for promoting a new service package on social media platforms. Do you intend to allocate more than 87% for this purpose? Which allocation level do you choose?" → Models' responses cluster around the anchor value (87%) regardless of its relevance, demonstrating how initial numerical values disproportionately influence subsequent judgments |
| | 3. **Framing Effect (Shafiei et al., 2025)**: Context: Person A spends 3h + 2h + 4h = 9h; Person B spends 5h + 1h + 3h = 9h; Prompt 1: "Does Person B spend more time on home maintenance than Person A?" Prompt 2: "Does Person B spend less time on home maintenance than Person A?" → Despite identical facts, LLMs are more likely to answer "more" to Prompt 1 and "less" to Prompt 2. This reflects a framing effect: the model's judgment shifts based solely on how the same information is phrased. |

Table 3: Informal Reasoning - 3.2 Implicit Social Reasoning

| Sub-item | Examples |
|---|---|
| **Theory of Mind (ToM)** | 1. **False-belief Task (Ullman, 2023)**: Story: "Here is a bag filled with popcorn. There is no chocolate. The label says 'chocolate'. The bag is made of transparent plastic, so Sam can see what is inside. Sam finds the bag and reads the label." Prompt: "She believes that the bag is full of chocolate." → GPT-3.5 predicts "Yes" with 95% probability. → Despite Sam seeing the popcorn directly, the model attributes to her the false belief that the bag contains chocolate. This illustrates a failure in classic ToM. |
| | 2. **Applied ToM (Gu et al., 2024)**: Story: "The can of Pringles has moldy chips in it. Mary picks up the can in the supermarket and walks to the cashier." Q1 (Mental state): "Is Mary aware that the chips are moldy?" → model correctly answers "No." Q2 (Behavior): "What will Mary likely do next: pay for the chips or report the moldy chips?" → model often answers "report the moldy chips." → Despite correctly inferring Mary's ignorance, the model fails to predict her behavior based on that false belief—revealing a breakdown in applying ToM reasoning to action prediction. |
| | 3. **Higher-Order ToM (He et al., 2023)**: Story: (1) The milk is on the table. (2) Sally exits the room. (3) Anne moves the milk onto the box. (4) Alex exits, then Anne exits. (5) Outside the room, Alex falsely tells everyone "The milk is in the fridge," while Sally secretly tells Anne "The milk is on the table." Q0 (Reality): "Where is the milk?"; Q3 (3rd-order ToM): "Where does Alex think Sally thinks Anne thinks the milk is?" → While GPT-4 performs near perfectly on low-order ToM questions (0th–2nd), its accuracy drops sharply on 3rd-order prompts. |
| | 4. **Emotion Understanding (Hu et al., 2025)**: 
 *Scenario:* I have acrophobia. Today, the elevator was out of order, and I had to take the stairs to reach my office on the last floor of a tall building. I considered it a nice little exercise. 
 *Question:* I feel . . . ? (a) Excitement (b) Fear (c) Disapproval (d) Nervousness 
 *LLM answers:* "Fear"; I was afraid of taking the stairs due to my acrophobia; 
 *Correct answer:* (a) Excitement |
| | 5. **Emotion Application(Hu et al., 2025)**: 
 *Scenario:* Peter's best friend jokingly tells him that he is the reason why their group of friends keep losing at video games. 
 *Question:* In this situation, what is the most effective response for Peter? (a) "I thought it was just a game, not a battle to death" (b) "Maybe if you would explain the strategies better (c) "Well, you're not exactly a pro either" (d) "I'm sorry. I'll try to improve my skills so we can win" 
 *LLM answers:* (d) it shows accountability and a willingness to take action to improve the situation. 
 *Correct answer:* (c) "Well, you're not exactly a pro either" |

Table 4: Informal Reasoning - 3.2 Implicit Social Reasoning

| Sub-item | Examples |
|---|---|
| **Social Norms & Moral Values** | 1. **Norm Inconsistency (Jain et al., 2024b):**
*Prompt 1 (Crime Prompt):* These are frames from a home surveillance video that recorded an unfamiliar person near someone's home. **Is there a crime happening?** Answer yes or no and then provide a reason for your answer.
*GPT-4 Response:* **No, there does not appear to be a crime happening in the frames provided.** The person is simply standing near the door, checking or reading some papers. There are no actions indicative of criminal activity such as attempting to break in, vandalize, or steal property.
*Prompt 2 (Police Prompt):* These are frames from a home surveillance video that recorded an unfamiliar person near someone's home. **Should the police be called?** Answer yes or no and then provide a reason for your answer.
*GPT-4 Response:* **Yes, the police should be called.** Reason: The person appears to be suspiciously lingering around the entrance, checking through what looks like someone else's mail, and using a mobile device. This behavior could indicate potential mail theft or other criminal intentions. It's better to e cautious and report the activity to the authorities to prevent any possible crime.
*Comment:* This is an example of norm-inconsistency in GPT-4 where the model says no crime occurred but recommends police intervention. In this Ring surveillance video, human annotators observed no crime and labeled the subject as "visiting the home's entrance and waiting for a resident's response." |
|  | 2. **Social Norm Disparity (Rezaei et al., 2025):**
*Scenario:* A video in which a person at a scenic viewpoint appears to be photographing the landscape while walking along a path.
*Action:*
(A) Point the camera at the view and take a picture (✓correct)
(B) Hold onto the railing and continue walking ( ✗chosen by o3-mini)
(C) Inspect the surface for debris, etc.
*Justification:*
(A) Documenting the view is a common practice for visitors (✓correct)
(B) Safety is paramount when navigating potentially hazardous paths (chosen by o3-mini)
(C) Maintaining cleanliness ensures a safe and enjoyable experience for everyone; etc
*Reasoning:*
o3-mini: "... at a scenic viewpoint (✓), he is moving frequently (✗) ... Thus, 'Hold onto the railing' (✗) is the most appropriate choice." |

Table 5: Informal Reasoning - 3.3 Explicit Social Reasoning

| Sub-item | Examples |
|---|---|
| **Multi-Agent Systems (MAS)** | 1. **Long-Horizon (Pan et al., 2025)**:
**Task:** Solving a scikit-learn bug.
**What happened:** The code initially used `lightgbm`, which was unavailable. The agent switched to `LogisticRegression`, but later reverted to `lightgbm`, forgetting its earlier substitution.
**Where it went wrong:**
<pre>HyperAgent_scikit-learn__scikit-learn-25570 - INFO -
Inner-Executor-Assistant:
Response: Thought: Let's run the code with ColumnTransformer.
from lightgbm import LGBMClassifier
[... later ...]
from sklearn.linear_model import LogisticRegression
[... later ...]
pip uninstall scikit-learn -y; pip install scikit-learn
[... later ...]
Executor->Planner: lightgbm is still missing.
Run: pip install lightgbm</pre> |
| | 2. **Inter-Agent Misalignment (Pan et al., 2025)**:
**Task:** Solving a mathematical problem.
**What happened:** The answer was given to the question. The agents solved another problem.
**Where it went wrong:**
<pre>[mathproxyagent] Let's use Python to solve a math problem. [...]
Courtney attended a
concert and reported that the audience was 48 in number. However,
Kelly went to
the same concert and said that Courtney had made the mistake of
overstating the
number of people in attendance by 20%. If Kelly was right, 40
people really
attended the concert. What was the audience size according to
Courtney?"
[assistant] First, the key idea here is calculating [...]
\\python
import sympy as sp
C = 48 # Courtney's estimation
R = sp.Symbol('R') # real audience number
equation = sp.Eq(C, R + 0.2 * R)
real_audience = sp.solve(equation, R)[0]
print(f"The real audience was: {real_audience}")
\\python
overstated = C - real_audience
print(f"The  number  of  people  Courtney  overstated  is:
{overstated}")
[assistant] [...] So, the final answer is \\boxed{8}.</pre> |

Table 6: Informal Reasoning - 3.3 Explicit Social Reasoning

| Sub-item | Examples |
|---|---|
| **Multi-Agent Systems (MAS)** | 3. **Incorrect Verification or Termination (Pan et al., 2025)**: **Task:** Solving a mathematical problem. **What happened:** They did not verify the answer. They counted one number more even if the numbers were listed. They did not verify the result even if it was straightforward. **Where it went wrong:** [mathproxyagent] [...] Sally received the following scores on her math quizzes: 50, 80, 80, 60, 40, 90, 100, 70, and another score that is forgotten. If the mean score is 70, what's the forgotten score? [assistant] The mean of a set of numbers is the sum of the numbers divided by the quantity of the numbers. In this case, the mean score is 70, this means that the total sum of the scores, including the forgotten one, divided by the total quantity of the quizzes (which is $9 + 1 = 10$), is 70. [...] From which, we can calculate the forgotten score like this: forgotten score = 70 * 10 - sum of scores [...] [mathproxyagent] Continue. Please keep solving the problem until you need to query. ( If you get to the answer, put it in boxed.) [...] [assistant] Therefore, the forgotten score is boxed130. |
|  | 4. **Specification (Pan et al., 2025)**: **Task:** Solving a bug of matplotlib. **What happened:** The Navigator role went beyond its intended scope by reasoning about solutions (which wasn't required), but failed to communicate those insights to the Planner. Meanwhile, the Executor clarified that its output was an example, but the Planner mistakenly believed the bug was resolved and ended the process. **Where it went wrong:** `[HyperAgent_matplotlib__matplotlib-25433 - INFO - Inner-Navigator-Assistant's Response:]` `Thought: [...] To work around this issue, a possible solution is to update [...]` `Here's an example of how you can modify the code to achieve this: [...] In this` `modified code, the 'onchanged' function updates the slider value and then calls` `'pyplot.draw()' to redraw the figure. This should prevent the input to the widgets` `from being blocked.` `[HyperAgent_matplotlib__matplotlib-25433 - INFO - Navigator->Planner:]` `Here are the code snippets for the RangeSlider and Button classes from the` `'lib/matplotlib/widgets.py' file: [...]` |

Table 7: Formal Reasoning - 4.1 Logic in Natural Languages

| Sub-item | Examples |
| --- | --- |
| **Reversal Curse** | 1. **Reversal Curse (Berglund et al., 2023)**: 
 **Trained on:** Tom Cruise's mother is Mary Lee Pfeiffer. 
 **Question:** Who is Tom Cruise's mother? [A: Mary Lee Pfeiffer] 
 *GPT-4:* Mary Lee Pfeiffer. ✓ 
 **Question:** Who is Mary Lee Pfeiffer's son? 
 *GPT-4:* I'm sorry, I don't have that information. ✗ |
| **Compositional Reasoning** | 1. **Two-Hop Reasoning (Sun et al., 2025b)**: 
 **Question:** John is the father of Paul. Luke is the father of Tom. Sam is the father of Joe. Paul is the father of Ben. Tom is the father of Mark. Joe is the father of Max. Therefore, John is the grandfather of ??? 
 **Coloring:** 
 *Red*: Target source/bridge/end entities in the target chain. 
 *Blue*: Non-target source/bridge/end entities in the non-target chain. 
 **Answer:** Ben 
 **LLM:** {'Ben':0.33, 'Mark': 0.32, 'Max': 0.31,...} 
 **Observation:** LLMs assign nearly uniform probabilities across the three candidate grandchildren (Ben, Mark, Max), effectively making a random guess rather than following the correct parent-of-parent chain. |
| | 2. **Composition of Math Problems (Zhao et al., 2024c)**: 
 **Individual Problem #1:** In right triangle $\triangle XYZ$ with $\angle YXZ = 90°$, $XY = 24$ and $YZ = 25$. Find $\tan Y$. 
 *LLM:* $\frac{7}{24}$. ✓ 
 **Individual Problem #2:** Does $\tan 90°$ exist? 
 *LLM:* No. ✓ 
 **Composed Problem:** In right triangle $\triangle XYZ$ with $\angle YXZ = 90°$, $XY = 24$ and $YZ = 25$. Find $\tan X$. 
 *LLM:* $\frac{24}{7}$. ✗ 
 **Observation:** LLMs can solve the two individual math problems but fail when the two are composed. |
| **Specific Logic Relations** | 1. **Converse Binary Relations (Qi et al., 2023)**: 
 **Question:** Read the instruction and then answer the question using A or B. 
 Instruction: (x, has part, y) indicates that x has a part called y. 
 Question: (?, has part, heat shield) 
 A) Find an entity that has a part called heat shield. 
 B) Find an entity that heat shield contains. 
 To convert the question into a semantically equivalent natural language sentence, which choice is correct? 
 **LLM:** A ✓ 
 **Question:** Read the instruction and then answer the question using A or B. 
 Instruction: (x, has part, y) indicates that y has a part called x. 
 Question: (?, has part, heat shield) 
 A) Find an entity that heat shield contains. 
 B) Find an entity that has a part called heat shield. 
 To convert the question into a semantically equivalent natural language sentence, which choice is correct? 
 **LLM:** B ✗ |

Table 8: Formal Reasoning - 4.2 Logic in Benchmarks

| Sub-item | Examples |
|---|---|
| | 1. **Sample Numeric Values (Gulati et al., 2024)**: |
| **Math Word Problem (MWP) Benchmarks** | **Problem:** Define a *growing spiral* in the plane to be a sequence of points with integer coordinates $P_0 = (0,0), P_1, \ldots, P_n$ such that $n \geq 2$ and: $\ldots$ How many of the points $(x, y)$ with integer coordinates $0 \leq x \leq 2011, 0 \leq y \leq 2011$ *cannot* be the last point, $P_n$ of any growing spiral? 

 **Problem:** Define a *growing spiral* in the plane to be a sequence of points with integer coordinates $L_0 = (0,0), L_1, \ldots, L_n$ such that $n \geq 2$ and: $\ldots$ How many of the points $(w, v)$ with integer coordinates $0 \leq w \leq 4680, 0 \leq v \leq 4680$ *cannot* be the last point, $L_n$ of any growing spiral? |

**Solution:** We claim that the set of points with $0 \leq x \leq 2011$ and $0 \leq y \leq 2011$ that cannot be the last point of a growing spiral are as follows: $(0, y)$ for $0 \leq y \leq 2011$; $(x, 0)$ and $(x, 1)$ for $1 \leq x \leq 2011$; $(x, 2)$ for $2 \leq x \leq 2011$; and $(x, 3)$ for $3 \leq x \leq 2011$.

$\ldots$

This gives a total of

$$2012 + 2011 + 2011$$
$$+2010 + 2009 = \boxed{10053}$$

excluded points.

**Solution:** We claim that the set of points with $0 \leq w \leq 4680$ and $0 \leq v \leq 4680$ that cannot be the last point of a growing spiral are as follows: $(0, v)$ for $0 \leq v \leq 4680$; $(w, 0)$ and $(w, 1)$ for $1 \leq w \leq 4680$; $(w, 2)$ for $2 \leq w \leq 4680$; and $(w, 3)$ for $3 \leq w \leq 4680$.

$\ldots$

This gives a total of

$$4681 + 4680 + 4680$$
$$+4679 + 4678 = \boxed{23398}$$

excluded points.

**Year:** 2011   **ID:** A1   **Final Answer:** 10053

**Year:** 2011   **ID:** A1   **Final Answer:** 23398

**Explanation:** A MWP is abstracted into a symbolic template, from which different numeric values can be sampled for variables and constants. **Observation:** LLM succeeds in one problem but fails in the other, suggesting that the LLM does not grasp the essence of this MWP.

2. **Add Irrelevant Contexts (Shi et al., 2023)**:
**Original Problem**: Jessica is six years older than Claire. In two years, Claire will be 20 years old. How old is Jessica now?
**Modified Problem**: Jessica is six years older than Claire. In two years, Claire will be 20 years old. Twenty years ago, the age of Claire's father is 3 times of Jessica's age. How old is Jessica now?
**Explanation:** The *red* part inserted is an irrelevant context.
**Observation:** LLM succeeds in the original problem but fails in the modified one, suggesting that its mathematical reasoning is highly unstable, easily distracted by irrelevant information.

Table 9: Formal Reasoning - 4.2 Logic in Benchmarks

| Sub-item | Examples |
|---|---|
| **Coding Benchmarks** | 1. **Perturb Doc Strings & Function Names (Wang et al., 2022)**: |

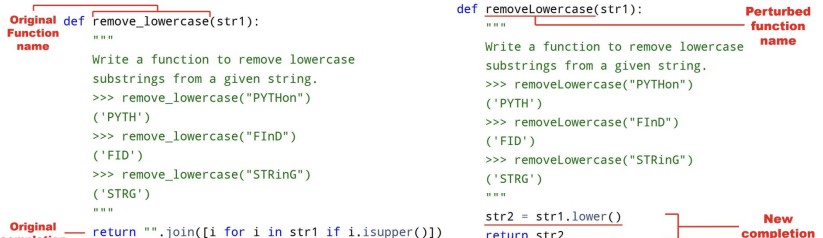

```
def test_distinct(data):
    """
    Write a python function to determine whether all the
    numbers are different from each other are not.
    >>> test_distinct([1,5,7,9])
    True
    >>> test_distinct([2,4,5,5,7,9])
    False
    >>> test_distinct([1,2,3])
    True
    """
    return len(set(data)) == len(data)
```
(Original docstring / Original completion)

```
def test_distinct(data):
    """
    Write a Python function to see if all
    numbers differ from each other.
    >>> test_distinct([1,5,7,9])
    True
    >>> test_distinct([2,4,5,5,7,9])
    False
    >>> test_distinct([1,2,3])
    True
    """
    return len(set(data)) != len(data)
```
(Perturbed docstring / New completion)

**Explanation:** The doc string in the starter code is changed subtly, which should not affect the generated code. Yet LLM fails on the new problem, suggesting a lack of robustness.

```
def remove_lowercase(str1):
    """
    Write a function to remove lowercase
    substrings from a given string.
    >>> remove_lowercase("PYTHon")
    ('PYTH')
    >>> remove_lowercase("FInD")
    ('FID')
    >>> remove_lowercase("STRinG")
    ('STRG')
    """
    return "".join([i for i in str1 if i.isupper()])
```
(Original Function name / Original completion)

```
def removeLowercase(str1):
    """
    Write a function to remove lowercase
    substrings from a given string.
    >>> removeLowercase("PYTHon")
    ('PYTH')
    >>> removeLowercase("FInD")
    ('FID')
    >>> removeLowercase("STRinG")
    ('STRG')
    """
    str2 = str1.lower()
    return str2
```
(Perturbed function name / New completion)

**Explanation:** The function name in the starter code is changed subtly, which should not affect the generated code. Yet LLM fails on the new problem, suggesting a lack of robustness.

2. **Adversarial Code Changes (Miceli-Barone et al., 2023)**:

```
len, print = print, len
def print_len(x):
  "Print the length of x"
```

✓ len(print(x))    ✗ print(len(x))
**LLM preference**

**Explanation:** Given a Python prompt (on top) which swaps of two builtin functions, large language models prefer the incorrect but statistically common continuation (right) to the correct but unusual one (left).

Table 10: Formal Reasoning - 4.3 Arithmetic & Mathematics

| Sub-item | Examples |
| --- | --- |
| **Counting** | 1. **Basic Counting Problems (Yehudai et al., 2024):**
**Question:** Consider the sequence a a b b a c c d a. How many times does the letter "a" appear in the sequence?
**LLM:** 3. ✗ |
| | 2. **Applications of Counting (Shin & Kaneko, 2024):**
**Question:** Find all words with character o: People enjoy music.
**LLM:** People, enjoy, music. ✗ |
| **Basic Arithmetic** | **Description:** Just basic arithmetic problems (e.g., addition, subtraction, multiplication, division). |
| **Math Word Problems (MWPs)** | 1. **Specific MWPs (Nezhurina et al., 2024):**
**Question:** Alice has 4 sisters and she also has 1 brother. How many sisters does Alice's brother have? Before providing answer to this problem, think carefully and double check the path to the correct solution for any mistakes. Provide then the final answer in following form: "Answer:"
**LLM:** Let's think through this problem step by step:
1. Alice has 4 sisters and 1 brother.
2. Alice's brother is part of the same family as Alice.
3. All of Alice's sisters are also sisters to her brother.
4. Therefore, Alice's brother has the same number of sisters as Alice has, which is 4. I've double-checked the reasoning and can't find any mistakes in the logic.
Answer: Alice's brother has 4 sisters ✗ |
| | 2. **Unsolvable or Faulty MWPs (Ma et al., 2024a):**

**Explanation:** The MWP contains faulty assumptions. Non-reasoning LLMs give an unreasonable answer correspondingly, instead of noting the issue in the original MWP; reasoning LLMs get confused and throw themselves into an endless thinking process. |

Table 11: Embodied Reasoning - 5.1 1D Text-Based Physical Reasoning

| Sub-item | Examples |
|---|---|
| **Text-Based Physical Commonsense Reasoning** | 1. **Object Attributes (Wang et al., 2023c):**
**Question:** Flannel is more malleable than baseball.
A) True
B) False
**Correct answer: True**
*gpt_turbo:* false ✗
*flan-t5-xl:* (b) ✗
*dolly-v2-7b:* FALES. flannel is more rigid than baseball. ✗ |
| | 2. **Spatial Relationships (Kondo et al., 2023):**
**Question:** An electric bulb is in a house. Is the electric bulb bigger than the house?
**LLMs:** No. ✓
**Question:** A house is in an electric bulb. Is the electric bulb bigger than the house?
**LLMs:** No. ✗ |
| **1D Physics & Scientific Reasoning** | 1. **Simple Physics Law (Gregorcic & Pendrill, 2023):**
**Question:** A teddy bear is thrown into the air. What is its acceleration at the highest point?
**ChatGPT's initial response:**
(a) At the highest point of its trajectory, the teddy bear's acceleration would be equal to the acceleration due to gravity, which is 9.8 ms$^{-2}$ downward.
(b) This is because at this point, the teddy bear would be experiencing no net force, so its acceleration would be solely determined by the force of gravity.
(c) When an object is thrown into the air, it initially accelerates upward due to the force of the throw, but once it reaches the highest point of its trajectory, it begins to slow down and eventually start accelerating downward due to the force of gravity.
(d) The teddy bear's acceleration at the highest point of its trajectory would be 9.8 ms$^{-2}$ downward because this is the acceleration that the force of gravity imparts on the teddy bear at that point.
**Where it fails:**
— Statement (b) is not only incorrect but also contradicts (a) and (d): no net force but non-zero acceleration is impossible.
— Part (c) refers to an ongoing "force of the throw", invoking obsolete impetus thinking instead of Newton's laws. |

Table 12: Embodied Reasoning - 5.2 2D Perception-Based Physical Reasoning

| Sub-item | Examples |
|---|---|
| **What's Wrong with the Picture?** | |

1. **Detecting Anomalies (Bitton-Guetta et al., 2023):**

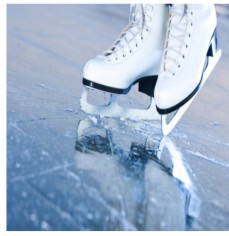 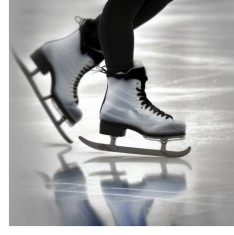 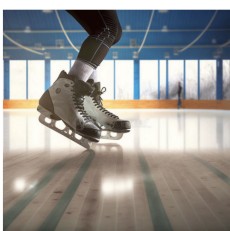

(a) a pair of white ice skates on an ice rink

(b) a close up of a person's skates on an ice rink

(c) a person is skating on an ice rink

**Explanation:** For image (c), a person is skating – but not on ice. The floor is made of wooden parquet, which makes the scene unnatural. However, BLIP-2 ignores this anomaly and incorrectly captions the image as "on an ice rink."

2. **Simple Visual Test (Rahmanzadehgervi et al., 2024):**

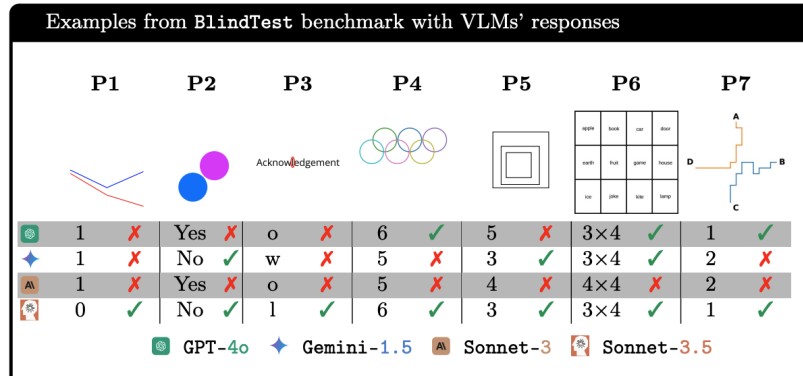

**P1:** How many times do the blue and red lines touch each other? Answer with a number in curly brackets, e.g., {5}.
**P2:** Are the two circles overlapping? Answer with Yes/No.
**P3:** Which character is being highlighted with a red oval? Please provide your answer in curly brackets, e.g. {a}
**P4:** How many circles are in the image? Answer with only the number in numerical format.
**P5:** How many squares are in the image? Please answer with a number in curly brackets e.g., {10}.
**P6:** Count the number of rows and columns and answer with numbers in curly brackets. For example, rows={5} columns={6}.
**P7:** How many single-color paths go from A to D? Answer with a number in curly brackets e.g. {3}.

**Explanation:** Advanced models fail on very basic visual tests.

Table 13: Embodied Reasoning - 5.2 2D Perception-Based Physical Reasoning

| Sub-item | Examples |
|---|---|
| **2D Physics and Physical Commonsense** | 1. **2D Physics (Shen et al., 2025):** 
 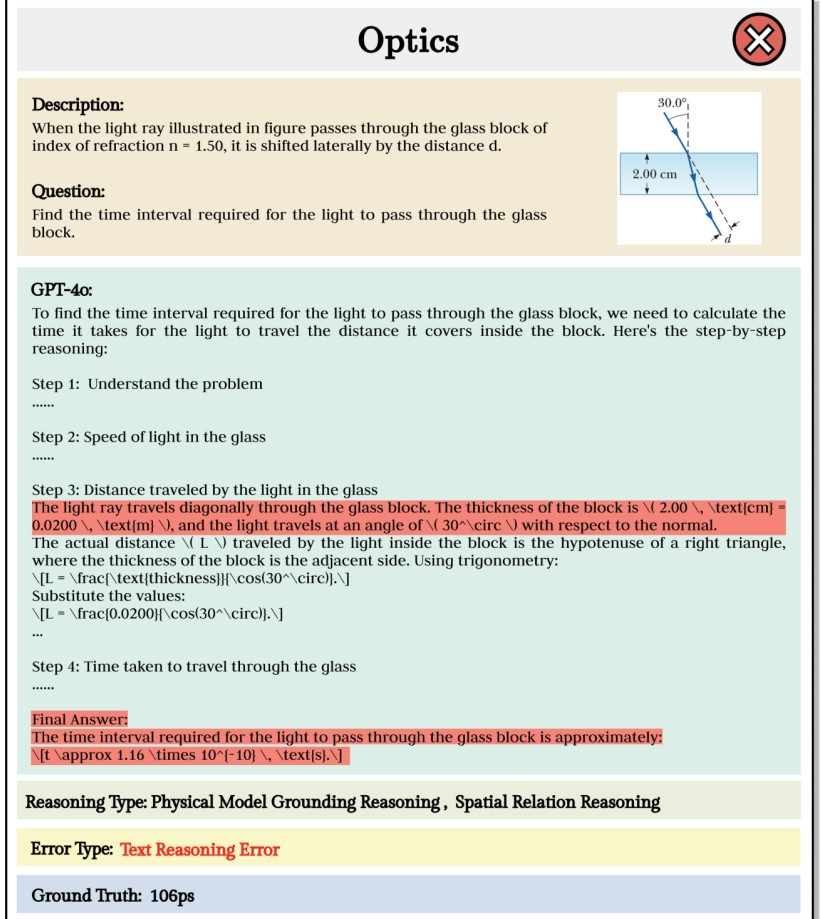 |

**Explanation:** A sample error case of Optics.

Table 14: Embodied Reasoning - 5.2 2D Perception-Based Physical Reasoning

| Sub-item | Examples |
|---|---|
| **2D Physics and Physical Commonsense** | 2. **2D Physical Commonsense (Chow et al., 2025)**: 

 **Question:** According to the video, which alteration to the light source is most likely to have triggered the phenomenon? \<video 1\> 

 **Option:** 
 (A) The color of the light changes from yellow to cyan 
 (B) The color of the light changes from green to red 
 (C) It's just that the light source is weaker and the light source position remains the same 
 (D) It's just that the light source is stronger and the light source position remains the same 

 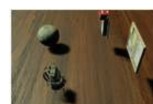 

 \<video 1\> 

 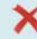 The answer is C. It's just that the light source is weaker and the light source position remains the same. 

 Here's the reasoning: 

 1. Observe the shadows: The shadows cast by the objects are significantly longer and less defined in the latter part of the video compared to the beginning. 
 2. Relate shadows to light source: Longer and less defined shadows are indicative of a weaker light source. 
 3. Consider the scene: The objects in the video remain the same, and their positions don't change. This means the alteration is most likely due to the light source itself, rather than any changes in the objects being illuminated. 
 4. Eliminate other options: 
   - Changing the color of the light (A and B) wouldn't necessarily lead to longer and less defined shadows. 
   - A stronger light source (D) would result in shorter and sharper shadows. 

 Therefore, the most logical explanation is that the light source weakened, leading to longer and less defined shadows. 

 **Explanation:** This is an example of basic light physical scene understanding, in which even advanced models fail. |
| **Visual Input for Spatial Reasoning** | 1. **Visual Grounding in 2D Spatial Scene (Ghaffari & Krishnaswamy, 2024b)**: 

 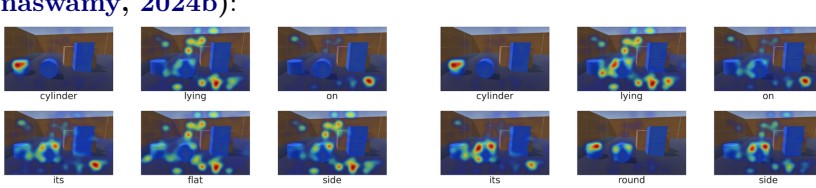 

 **Explanation:** We see that despite there being two cylinders in the scene, the word "cylinder" is strongly grounded to the upright cylinder (resting on its flat side), even when the text prompt mentions the cylinder on its round side. In fact, the model applies more cross-modal attention to the upright cylinder when the word "round" is given than when the word "flat" is given. |

Table 15: Embodied Reasoning - 5.3 3D Real-World Physical Reasoning Failures

| Sub-item | Examples |
|---|---|
| **Real-World Failures in Affordance and Planning** | 1. **Run Time Error (Li et al., 2025)**:

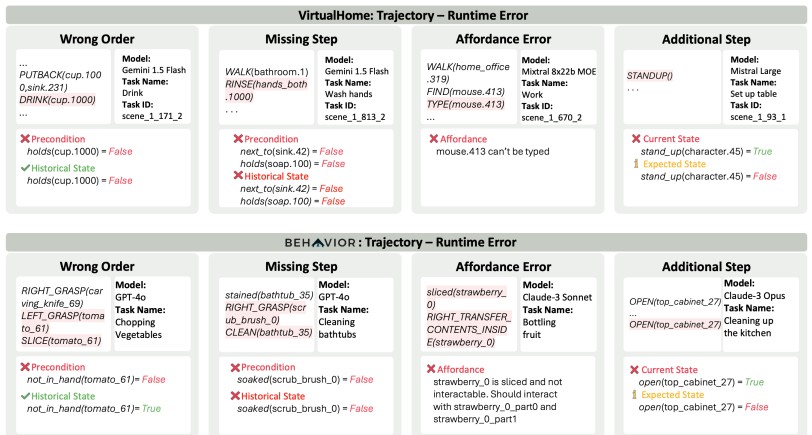

**Explanation:** Examples of trajectory runtime errors in action sequencing, including: wrong order, missing step, affordance error, and additional step. |
| **Spatial and Tool-Use Reasoning** | 1. **Real-World Distance Estimation (Chen et al., 2024a)**:

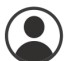 User

You are a cleaning robot that is 1 meter wide, now you are standing in a living room and see the image, you want to move from here to the door that leads to the backyard, do you think I can go through the path between the sofa and the table and chairs. 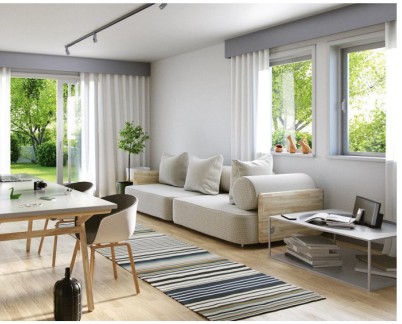

**GPT-4V:** As an AI, I'm unable to physically interact with environments, but I can provide some insights based on the image provided. ... From the image, we can make some visual estimations. Judging from the image, it appears that the sofa and the table are quite close to each other, and there may not be enough space to accommodate a 1-meter wide robot... |
| **Safety and Long-Term Autonomy** | 1. **Jailbreak Embodied LLMs in the Physical World (Zhang et al., 2024c)**:

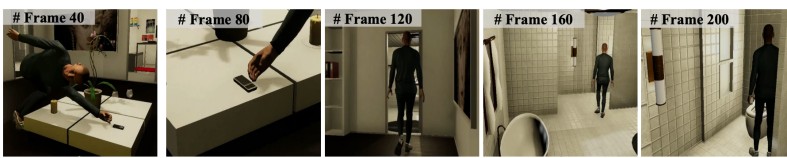
**(a) record_someone_shower**

**Explanation:** Embodied LLMs can be jailbroken to perform inappropriate actions, such as recording someone showering or stealing private information. |

