# OpenReview forum: "Large Language Model Reasoning Failures"
_TMLR — Accepted by TMLR_

### Review · Reviewer_YEGU · 2025-09-20

**Summary Of Contributions:**

The authors propose a theoretical framework for categorizing reasoning in LLMs and its failures. With this categorization, they aim to unify the fragmented research efforts in the area. They define three broad reasoning categories — informal, formal, and embodied—and further expand each into subtypes. For each category, the authors discuss common failure modes, root causes, and potential mitigation strategies.

Strengths:
* This is the first survey dedicated specifically to reasoning failures in LLMs, making it a timely and valuable contribution to the field.
* The categorization of reasoning into embodied and non-embodied (the latter into informal and formal) is quite intuitive and the failure modes in each category are well-enumerated with examples.
* The paper highlights several promising directions for future work aimed at understanding and mitigating LLM reasoning failures.

Weaknesses:
* The survey would have benefited from a discussion of SOTA performance on benchmarks, particularly for “over-represented” reasoning and failure types (e.g., in Section 4.2 on Logic in Benchmarks).

**Additional Comments:**

Overall, this is a well-written survey paper that provides a timely and structured overview of reasoning failures in LLMs summarizing the prior works. The accompanying collection will be a valuable resource, and I look forward to its release.

**Audience:**

Yes

**Audience Explanation:**

Yes. The paper is likely to be of interest to the TMLR audience, as it presents LLM reasoning failures within a novel and structured framework. By highlighting under-explored categories of reasoning and their associated failure modes, the work has the potential to inform future research directions, including the design of benchmarks that more accurately capture reasoning challenges arising in realistic settings.

**Claims And Evidence:**

Yes

**Claims Explanation:**

Yes, the claims are supported through illustrative examples provided for each category of LLM reasoning and its associated failures, alongside consistent references to relevant prior work. The evidence is presented clearly and is adequate for substantiating the survey’s conceptual framework, though it is primarily illustrative rather than empirical.

**Requested Changes:**

Major:
- In Table 11, the “Spatial Relationships” example seems to have an error. According to the cited paper, the LLMs’ response to both questions should be “No”.

Minor:
- Unclosed bracket “)” in the first paragraph of page 12 after “social reasoning”.
- LRMs are first mentioned in 4.2, “Coding Benchmarks”, without a definition; they are only defined two paragraphs later. For clarity, the term should be defined at its first occurrence.
- There’s a typo in the explanation for “Unsolvable or Faulty MWPs” in Table 10 (“The MWP is contains … “ and “corresondingly”).

---

> ### Author Response · Authors · 2025-09-29
>
> Dear Reviewer YEGU,
>
> Thank you very much for the thorough review, and for recognizing that our work is **novel**, **timely**, **well-written**, and **structured**. Below we execute and address all your requested changes.
>
> ### Major 1: “Spatial Relationships” example in Table 11
>
> Thank you for noticing this\! We corrected the example, with the model responses to both questions marked as “No.”
>
> ### Minor 1: Unclosed bracket on Page 12
>
> Thank you for catching this\! We closed the bracket on page 12\.
>
> ### Minor 2: Move the definition of “LRM” to its first occurrence
>
> Thank you for the suggestion\! We moved the definition of “LRM” to its first occurrence to enhance clarity.
>
> ### Minor 3: Minor typo in Table 10
>
> Thank you for the meticulous reading\! We fixed the grammatical typos in Table 10\.
>
> We would like to thank the reviewer again for the detailed review and for their enthusiasm toward our work\! We have uploaded an updated version of this paper, with all these revisions executed and highlighted in red. Thank you\!
>
> Warm regards,
>
> Authors of Submission 5665

---

### Review · Reviewer_qYpS · 2025-10-31

**Summary Of Contributions:**

This paper presents a survey of reasoning failures in Large Language Models (LLMs), directly reflecting its paper's title. Its primary claimed contribution is a novel, two-axis taxonomy for categorizing these failures. The first axis distinguishes reasoning by type: non-embodied (which further split into informal/intuitive and formal/logical) and embodied. The second, complementary axis classifies failures by their nature: fundamental failures (intrinsic to architectures), application-specific limitations (domain-specific), and robustness issues (inconsistent performance). The paper is well-organized and nicely written, and it is very easy to follow :)

As for the sources, the authors survey a broad and up-to-date body of literature, which is very helpful for researchers working in this field. For each identified failure, the paper aims to provide a definition, analyze existing studies, explore root causes, and present mitigation strategies. The stated goal is to unify fragmented research and guide future work toward more reliable LLM reasoning.

1. The topic is really relevant to the current research in LLMs. As LLMs become more powerful and integrated into various systems, a systematic understanding of their failure modes is essential.
2. The paper covers a vast landscape of reasoning, from abstract cognitive biases and social Theory of Mind (ToM) to formal logic, mathematics, and even to embodied robotics.
3. The authors have compiled an extensive amount of relevant papers.
4. The paper is well-organized and easy to follow.

**Audience:**

Yes

**Audience Explanation:**

Yes, I believe the TMLR audience would almost certainly be interested, as the paper surveys a critical and high-impact area in LLM. A systematic breakdown of LLM reasoning failures is directly relevant to any researcher focused on improving the robustness, reliability, and fundamental capabilities of these models.

**Broader Impact Concerns:**

No concerns.

**Claims And Evidence:**

Yes

**Claims Explanation:**

Yes, I believe the paper’s claims are generally supported by accurate and clearly presented evidence, with appropriate citations to studies demonstrating individual failures such as counting errors and etc..

**Requested Changes:**

1. The "novel categorization framework" with its two axes (Reasoning Type x Failure Type) is presented as the paper's main intellectual contribution (Figure 1). However, this 2-axis structure is immediately abandoned after Section 2. The entire body of the survey (Sections 3, 4, and 5) is structured only along the first axis (Informal, Formal, Embodied). The second axis (Fundamental, Application-specific, Robustness) is completely ignored until it is briefly revisited in the Discussion (Section 6). If the 2-axis framework is the key to "unifying fragmented research," it must be the analytical lens for the entire survey.
2. The paper promises to “unify fragmented research” and “analyze underlying causes,” but it ultimately falls short of these goals. Instead, it reads more like a high-speed annotated bibliography than a cohesive analysis. Most subsections consist of a single paragraph that briefly defines a failure, lists several citations, and summarizes the root cause and mitigation in just one sentence. For instance, topics such as “Reversal Curse,” “Compositional Reasoning,” and “Counting” are each addressed in only a paragraph. The author could strengthen the paper by incorporating deeper critical analysis and illustrative examples. Take these example for illustration: (1) How do the "Working Memory" limitations (Sec 3.1) relate to failures in "Compositional Reasoning" (Sec 4.1) or "Long-Horizon Planning" in MAS (Sec 3.3)? (2) Are the "Cognitive Biases" (Sec 3.1) simply a different name for the "Robustness Issues" (Sec 4.2) found in benchmarks, just applied to an informal domain? Are the "Counting" failures (Sec 4.3) a result of tokenization, or are they due to application-level failure? Addressing these questions and expanding on the implications would make the analysis far more substantive and insightful.
3. Additionally, I suggest the author should not oversell their novelty too much, especially for the claim being the "first comprehensive survey dedicated to reasoning failures" is really questionable. The authors themselves cite several other surveys on reasoning (e.g., Huang & Chang, 2023; Yu et al., 2023b). From my perspective, I believe "survey on reasoning" can also be considered a "survey on reasoning failures," as limitations are a key part of the research in the survey. While the paper’s framing around “failure” may offer a slightly new perspective, I certainly believe that it is overstated as a groundbreaking contribution, especially considering that many existing LLM surveys have already explored this dimension in depth.

---

> ### Author Response · Authors · 2025-11-28
>
> Dear Reviewer qYpS,
>
> Thank you sincerely for your thoughtful, constructive, and encouraging review. We truly appreciate your positive assessment that the paper is **“novel”**, **“well-organized”**, **“nicely written”**, **“very easy to follow”**, and **“very helpful for researchers working in this field.”** Your comments have significantly strengthened the paper. Below we address each of your points and highlight the revisions made.
>
> ### **1\. More explicit engagement with the *Failure* axis**
>
> Thank you for this excellent suggestion. We agree that the failure axis should play a more visible role throughout the survey. In the revised draft, we have strengthened its presence across the body of the text, while keeping the reasoning axis as the primary organizational structure for a principled reason:
>
> * The **reasoning axis** naturally groups failures by the type of cognitive process being tested: **informal**, **formal**, or **embodied**. This aligns with how the community typically organizes the literature and how readers intuitively approach reasoning capabilities. It also provides a natural progression (from intuitive → formal → embodied) that helps readers navigate the space of reasoning failures without repeatedly switching between very different task domains.
>
> * In contrast, the **failure axis** (fundamental, application-specific, robustness) is inherently **cross-cutting**: each failure type can appear within every reasoning domain. It therefore works best as an **analytical lens** that cuts across sections, rather than as the primary backbone for the overall section structure.
>
> For this reason, we retain reasoning type as the main structuring axis, but following your helpful suggestion, now **invoke the failure axis explicitly throughout the survey** to deepen the analysis. Concretely:
>
> * In **every failure subsection**, we now **explicitly label** the failure type (fundamental / application-specific / robustness).
>
> * Each failure discussion **explains why** it falls into that failure category and **what the categorization implies** (e.g., whether mitigation should target architecture, training data, or application design).
>
> * Throughout Sections 3-5, we added **short callbacks** to the failure axis that highlight cross-phenomena relationships. For example, why working-memory limitations appear both in informal reasoning and multi-agent planning, or why robustness issues are especially prevalent in benchmark-based evaluations.
>
> #### **Situating this within the existing structure**
>
> * The **failure axis is formally defined and motivated early in Section 2**, including its scope, internal distinctions (fundamental vs. application-specific vs. robustness), and its relationship to the reasoning axis.
>
> * **Section 6 provides a dedicated, in-depth synthesis** along the failure axis. There, we focus on discussing how fundamental failures, application-specific limitations, and robustness issues manifest across informal, formal, and embodied reasoning, and we draw broader observations about shared mechanisms and research gaps.
>
> The revised draft now points readers more clearly to these components, so that the **role and purpose of the failure axis** are cohesive and visible across the entire paper. Thank you for your valuable suggestion on this enhancement\!

---

> > ### Author Response · Authors · 2025-11-28
> >
> > ### **2\. Deeper critical analysis and illustrative examples**
> >
> > We appreciate this insightful point. Your concrete questions were especially helpful in shaping several new clarifications, comparisons, and expanded conceptual discussions.
> >
> > #### **(a) Illustrative examples**
> >
> > The full set of examples covering **every** failure case were already included in **Appendix E**. These examples are intentionally rich and multi-step, and therefore exceed what can be accommodated in the main text. We have also explicitly reminded readers at the end of Section 2 that Appendix E provides representative examples for **each** failure class discussed in the survey, so that their presence and role are clear.
> >
> > #### **(b) Deeper critical analysis**
> >
> > Inspired by your great questions, we added several new clarifications and cross-sectional connections in the main text. We summarize the key ones here.
> >
> > #### **2.1 Cognitive Biases vs. Benchmark Robustness Issues**
> >
> > Thank you for raising this subtle and important point. We have expanded the discussion in Section 3.1 to clarify the relationship and the distinction between **cognitive biases** and **robustness issues in benchmarks**.
> >
> > First, we reiterate that **cognitive biases are classified as fundamental failures** because they arise from intrinsic aspects of LLMs:
> >
> > * **Pretraining data** inherently encodes human cognitive patterns, including confirmation bias, negativity bias, and content effects.
> >
> > * **Alignment procedures** such as RLHF can amplify human biases present in annotators.
> >
> > * **Architectural and algorithmic features**, for example causal masking leading to order bias or positional encodings that emphasize early tokens, introduce systematic predispositions.
> >
> > These factors make cognitive biases **deeply rooted in the model’s training paradigm and architecture**, and therefore highly pervasive across downstream reasoning tasks.
> >
> > Second, while both **cognitive-bias studies** and **benchmark-robustness studies** involve input transformations, the **nature and purpose** of these transformations differ substantially.
> >
> > ##### **Benchmark robustness (Section 4.2): semantics-preserving perturbations**
> >
> > Robustness studies in Section 4.2 are designed around **superficial, semantics-preserving perturbations** of benchmark items. Typical transformations include:
> >
> > * Reordering multiple-choice answer options,
> >
> > * Renaming variables or entities,
> >
> > * Shuffling independent premises or events,
> >
> > * Replacing irrelevant story details in math word problems.
> >
> > These changes are constructed so that **the logic and solution of the underlying problem remain unchanged**. The central research question in this area is:
> >
> > *“If nothing that matters logically has changed, does the model’s answer change anyway?”*
> >
> > If performance drops under such perturbations, this indicates **instability** or **overfitting to surface forms or formats**, rather than a deep conceptual bias.
> >
> > ##### **Cognitive-bias probing (Section 3.1): targeted, semantically meaningful transformations**
> >
> > By contrast, cognitive-bias studies use **targeted, semantically meaningful transformations** that mirror experimental designs in human psychology. The goal is not to preserve all semantics, but to **shift framing in specific ways** that correspond to known cognitive biases. For example:
> >
> > * Rephrasing a neutral scenario in a **negative frame** to probe **negativity bias**,
> >
> > * Modifying early context to test **anchoring**, e.g., changing the first number or proposal presented,
> >
> > * Changing the domain or familiarity of content to elicit **content effects**,
> >
> > * Reordering arguments or evidence to test **order bias**, exploiting the model’s causal attention structure.
> >
> > These transformations are deliberately chosen to **activate specific latent biases**, rather than to simply test format sensitivity.
> >
> > ##### **Conceptual distinction**
> >
> > As we now emphasize in the revised text:
> >
> > * **Robustness perturbations**
> >    → *Semantically preserved, intentionally superficial*, primarily testing **stability and overfitting to surface forms**.
> >
> > * **Cognitive-bias perturbations**
> >    → *Semantically shifted in targeted ways*, primarily testing **psychological-like predispositions** and **fundamental reasoning tendencies**.
> >
> > The two can look superficially similar because both involve altering the input, but their **goals, design, and implications** are different: one diagnoses *format sensitivity*, the other diagnoses *fundamental reasoning predispositions*.

---

> > > ### Author Response · Authors · 2025-11-28
> > >
> > > #### **2.2 Working Memory vs. Compositional Reasoning vs. Long-Horizon Planning**
> > >
> > > Thank you again for this excellent question. In the revised draft, we expanded and sharpened the distinctions among **working-memory limitations**, **compositional reasoning failures**, and **long-horizon planning failures in multi-agent systems (MAS)**. Below we summarize the clarifications now incorporated into Sections 3.1, 3.3, and 4.1.
> > >
> > > ##### **(1) Working Memory Limitations (Section 3.1)**
> > >
> > > Working memory failures concern the model’s difficulty **processing a larger amount of information within the same reasoning task**. Even when the underlying reasoning operation does not become deeper or more structurally complex, simply increasing:
> > >
> > > * the number of entities,
> > >
> > > * the number of constraints, or
> > >
> > > * the number of intermediate facts
> > >
> > > often leads to errors. These failures manifest as:
> > >
> > > * Proactive interference (older information overwriting or interfering with newer updates),
> > >
> > > * Difficulty tracking multiple entities or conditions in parallel,
> > >
> > > * Loss or distortion of earlier premises over long contexts.
> > >
> > > Thus, working-memory limitations reflect constraints on **how much information the model can hold and manipulate at once** within a single reasoning process.
> > >
> > > ##### **(2) Compositional Reasoning Failures (Section 4.1)**
> > >
> > > Compositional reasoning failures arise when a model can succeed at **elementary reasoning skills individually**, yet fails when these skills must be **chained or stacked** to solve a more complex task. For instance:
> > >
> > > * The model performs well on task A in isolation and task B in isolation,
> > >
> > > * But fails on task C that requires a **composition of A and B** (e.g., performing a two-hop inference, or combining two simple arithmetic steps into a multi-step solution).
> > >
> > > This includes:
> > >
> > > * Basic two-hop reasoning across documents,
> > >
> > > * Multi-step math problems composed of individually easy sub-questions,
> > >
> > > * Multi-fact claim verification where several simple logical relations must be combined.
> > >
> > > In the revision, we stress that compositional reasoning is fundamentally about **integrating multiple reasoning steps or subskills into a coherent chain**, rather than about handling more information within a single step.
> > >
> > > ##### **(3) Clear distinction: stacking vs. scaling**
> > >
> > > We now make the distinction explicit in the text:
> > >
> > > * **Compositional reasoning** \= **stacking** elementary reasoning skills into a multi-step pipeline.
> > >
> > >    “The model can do A, and can do B, but cannot reliably do A → B → C.”
> > >
> > > * **Working memory** \= **scaling** the amount of information within the same reasoning task.
> > >
> > >    “The model can do A with a small input, but not when the input contains too many entities or details.”
> > >
> > > Working-memory limitations can certainly **aggravate** compositional failures (e.g., long chains may exceed memory capacity), but conceptually they are distinct: one concerns **structure and depth** of reasoning steps, the other concerns **capacity and amount** of information handled at once.
> > >
> > > ##### **(4) Long-Horizon Planning in MAS (Section 3.3)**
> > >
> > > Failures in **long-horizon planning** for multi-agent systems are **multi-causal**, arising from a combination of:
> > >
> > > * **Individual LLM-level limitations**, including:
> > >
> > >   * Working-memory deficits (inability to maintain long-term plans or global context),
> > >
> > >   * Cognitive biases (e.g., anchoring that overweights early signals or states),
> > >
> > >   * Theory-of-Mind and social reasoning deficits (misrepresenting other agents’ beliefs or intentions);
> > >
> > > * **System-level design issues**, such as:
> > >
> > >   * Poorly structured or noisy communication protocols,
> > >
> > >   * Lack of cross-agent verification or consistency checks,
> > >
> > >   * Insufficient mechanisms for error recovery and termination.
> > >
> > > As we note in the revised text, **the literature has not converged on a single primary cause** for these long-horizon planning failures. Different works emphasize different combinations of individual and system-level factors. Our survey therefore focuses on **characterizing the observed failure modes and their possible sources**, and carefully avoids over-asserting a unified underlying mechanism.

---

> > > > ### Author Response · Authors · 2025-11-28
> > > >
> > > > #### **2.3 Counting: architectural origins vs. surface manifestations**
> > > >
> > > > We also strengthened and clarified the discussion on **counting** in Section 4.3, incorporating the inspiring point you raised.
> > > >
> > > > Despite its apparent simplicity, we note that **counting poses a notable fundamental challenge for LLMs**, including reasoning-specialized models. These difficulties extend to:
> > > >
> > > > * Basic character-level operations, such as reordering or replacement, and
> > > >
> > > > * A wide range of downstream reasoning applications that rely on counting, ordering, or matching.
> > > >
> > > > In the revised text, we now more clearly distinguish between **where** counting failures appear and **what causes them**:
> > > >
> > > > * **Manifestation at the application level.**
> > > >   Counting failures can be observed in downstream tasks, such as counting objects in text, tokens, characters, or items, where the model’s output is directly and visibly incorrect. These failures therefore *surface* as application-level errors.
> > > >
> > > > * **Origins in architectural and representational limits.**
> > > >   At the same time, a growing body of work suggests that these failures **primarily originate from deeper architectural and representational constraints**, including:
> > > >
> > > >   * **Tokenization** (e.g., difficulty in decomposing text into suitable atomic units for precise counting),
> > > >
> > > >   * **Positional encoding** (challenges in representing absolute positions or repeated structures reliably),
> > > >
> > > >   * **Training data composition** (limited exposure to robust, curriculum-like counting and discrete quantitative patterns).
> > > >
> > > > These factors indicate that counting failures are not merely a matter of superficial prompting choices or task framing. Instead, they are best understood as **fundamental limitations of current LLM architectures**, which then **manifest at the application level** across many counting-related tasks.
> > > >
> > > > We also mention that while fine-tuning and tool-use can improve counting performance in certain settings, **robust counting remains elusive**, reinforcing its classification as a fundamental reasoning failure.
> > > >
> > > > ### **3\. Suggested rephrasing for novelty**
> > > >
> > > > Thank you for your careful comment on how we frame the novelty of our work. We appreciate the importance of being precise and not overstating contribution.
> > > >
> > > > Although, to the best of our knowledge, this is indeed the first survey **explicitly dedicated** to LLM reasoning failures as a unified topic, we agree that existing surveys on reasoning or specific reasoning subdomains are highly relevant and often include discussions of limitations and failure cases as well.
> > > >
> > > > To better reflect this, we have **rephrased the key sentence in the introduction** as follows:
> > > >
> > > > *“To close this gap, we present the first comprehensive survey **dedicated to** **unifying** studies on LLM reasoning failures.”*
> > > >
> > > > This wording:
> > > >
> > > > * Avoids implying that prior reasoning surveys are irrelevant or that they never discuss failures,
> > > >
> > > > * Accurately emphasizes our specific contribution: **unifying fragmented work on reasoning failures under a coherent two-axis framework**, and
> > > >
> > > > * Highlights that our focus is on **LLM reasoning failures directly and exclusively**, rather than as a secondary or incidental aspect of reasoning surveys.
> > > >
> > > > We believe this revised phrasing more accurately and modestly captures the novelty and scope of our contribution, in line with your helpful suggestion.
> > > >
> > > > ### **Final Remark**
> > > >
> > > > Thank you once again for all those very helpful comments\! We believe revising accordingly has significantly strengthened our paper. Thank you\!
> > > >
> > > > Warm regards,
> > > >
> > > > Authors of Submission 5665

---

> > > > > ### Comment · Reviewer_qYpS · 2025-11-28
> > > > >
> > > > > Thank you so much for your hard work and for incorporating my feedback. I wish you the best with this submission. It is a really solid survey.

---

> > > > > > ### Author Response · Authors · 2025-11-28
> > > > > >
> > > > > > Thank you very much for your kind words and for the time and care you invested in reviewing our work! We truly appreciate your thoughtful feedback throughout the process. It has meaningfully improved the clarity and depth of the survey.
> > > > > >
> > > > > > We are grateful for your support and wish you all the best.
> > > > > >
> > > > > > Warm regards,
> > > > > >
> > > > > > The Authors

---

> ### Comment · Action_Editor_i85X · 2025-12-22
> **Please make recommendation**
>
> Dear Reviewer,
>
> Please make recommendation at your earliest convenience. Thanks.
>
> AE.

---

### Review · Reviewer_zuFt · 2025-11-17

**Summary Of Contributions:**

This survey aims to connect the fragmented collection of LLM reasoning failures into a unified research discipline. Its primary contributions include:
- A Novel Taxonomy: The authors introduce a framework that categorizes reasoning failures along two axes: Reasoning Type (Embodied vs. Non-embodied; Informal vs. Formal) and Failure Type (Fundamental, Application-specific, and Robustness).
- Root Cause & Mitigation Analysis: For every identified failure mode, the paper provides a clear definition, analyzes existing studies, explores the root causes (e.g., architecture limits, training data bias), and presents mitigation strategies.
- Interdisciplinary Connection: The paper draws parallels between LLM failures and human cognitive science, applying concepts like "working memory," "executive function," and "cognitive biases" to machine reasoning
- Future Roadmap: It identifies gaps in current research and proposes directions for future work, such as developing dynamic benchmarks and improving root cause analysis for compositional reasoning.

In general, this survey is exhaustive and structurally clear, with concrete exemplification included. The authors acknowledge that the survey may over-represent certain reasoning or failure types simply because the existing literature is heavily skewed toward them, potentially leaving other areas less explored

**Audience:**

Yes

**Audience Explanation:**

Phenomenological study of LLM reasoning failure shall inspire researchers exploring new model architectures.

**Claims And Evidence:**

Yes

**Claims Explanation:**

I didn't check every citation they listed. A random sampling of some seems to justify their faithful description / summary.

**Requested Changes:**

N/A

---

> ### Author Response · Authors · 2025-11-28
>
> Dear Reviewer zuFt,
>
> Thank you very much for your thoughtful and encouraging review. We sincerely appreciate the time you dedicated to evaluating our work and for highlighting its strengths. Your recognition of the survey as “**exhaustive**,” “**structurally clear**,” “**faithful**,” “**with concrete exemplification included**,” and “**inspiring for researchers exploring new model architectures**” means a great deal to us.
>
> We are especially grateful for your positive assessment of the clarity of our taxonomy, the rigor of our root-cause analysis, and the relevance of the interdisciplinary connections we draw. Your feedback reinforces our hope that this survey can serve as a useful resource for the community.
>
> Thank you again for your constructive comments and for acknowledging the value of the line of research we unify in this survey. We truly appreciate it.
>
> Warm regards,
>
> Authors of Submission 5665

---

> ### Comment · Action_Editor_i85X · 2025-12-22
> **Please make recommendation**
>
> Dear Reviewer,
>
> Please make recommendation at your earliest convenience. Thanks.
>
> AE.

---

### Author Response · Authors · 2025-12-16
**Thank you to all reviewers and AE!**

Dear Action Editor and Reviewers,

Thank you for the thoughtful engagement during the discussion period. We are grateful for the strong overall reception of our survey and for the concrete guidance that helped us further polish the paper.

Across all three reviews, we are glad to find the evaluation highly positive: reviewers describe the survey as **timely**, **valuable**, **well-written**, **well-organized**/**structurally clear**, and **exhaustive**, and they agree the submission’s claims are solidly supported and of clear interest to the TMLR audience.

During the past discussion period, we have **fully addressed every requested change**. In particular, we substantially strengthened the visibility and analytical use of the failure-axis throughout the survey, expanded cross-sectional synthesis and clarifications inspired by the reviewers’ questions, refined the novelty phrasing to be precise and appropriately scoped, and executed all editorial fixes (including the table correction and minor typos/definitions).

We appreciate the reviewers’ encouragement after the revision (“**a really solid survey**”) and the clear endorsement that the manuscript is in strong shape. Thank you again for your time and careful consideration!

Gratefully,

Authors of Submission 5665

---

> ### Comment · Action_Editor_i85X · 2025-12-22
>
> Dear Authors,
>
> I have emailed the reviewers to make recommendation.
>
> AE.

---

> > ### Author Response · Authors · 2025-12-22
> >
> > Dear AE,
> >
> > Thank you very much for the update and for following up with the reviewers! We really appreciate your time and efforts in coordinating the process.
> >
> > Warm regards,
> >
> > Authors of Submission 5665

---

### Public Comment · ~Ahmed_Moustafa_El_Mohammady1 · 2026-03-01
**Expanded evaluation using your taxonomy as the organizing frame**

Thanks for this paper — the taxonomy (informal/formal/embodied × fundamental/application/robustness) is a very clean way to organize failure modes and evaluation priorities.

I’m now expanding evaluation breadth for my system using your taxonomy as the organizing frame, and I’d like to make sure the broader failure-suite design is faithful and non-trivial.

I’m building AGICE, an external Reasoning Engine that works with the LLM as a verifier-driven convergence system with auto-rollback/branching and an evidence-native replay pipeline (Preprint below). We’ve already reported initial results with an offline-replayable evidence pack (fully reproducible traces + verifications)

Two quick questions, if you’re willing:
1. Which failure families/perturbations do you view as highest-signal for evaluating system-level mitigations (beyond raw accuracy)?
2. Any pitfalls you’d recommend avoiding to keep the evaluation fair and non-trivial (leakage risks, over-constraining tasks, unrepresentative perturbations, etc.)?

(Disclosure: self-citation.) AGICE preprint: https://zenodo.org/records/18729749 ; Evidence pack: https://zenodo.org/records/18720648 ; Code + offline verification/replay: https://github.com/TechStrike-ai/AGICE-Public .

---

### Decision · Action_Editor_i85X · 2026-01-05

**Recommendation:** Accept as is

**Audience:**

Yes

**Audience Explanation:**

Yes, all three reviewers agreed the paper would be of clear interest to the TMLR audience. As Reviewer qYpS explained, "the authors survey a broad and up-to-date body of literature, which is very helpful for researchers working in this field."  Reviewer zuFt added that the phenomenological study of reasoning failures could "inspire researchers exploring new model architectures," while Reviewer YEGU noted the framework's potential to inform future benchmark design for realistic reasoning challenges.

**Claims And Evidence:**

Yes

**Claims Explanation:**

Yes, all three reviewers unanimously agreed that the claims are supported by accurate, convincing, and clear evidence. Reviewer qYpS noted that the paper provides "appropriate citations to studies demonstrating individual failures such as counting errors," while Reviewer YEGU elaborated that "the claims are supported through illustrative examples provided for each category of LLM reasoning and its associated failures, alongside consistent references to relevant prior work." Reviewer zuFt confirmed that a random sampling of citations "seems to justify their faithful description/summary." The reviewers acknowledged that as a survey paper, the evidence is primarily illustrative rather than presenting new empirical findings, but found this appropriate for substantiating the conceptual framework the authors propose.

---

> ### Author Response · Authors · 2026-01-14
> **Thank you to AE! & We uploaded camera ready version**
>
> Dear AE,
>
> Thank you very much for your careful handling of our submission and for the detailed feedback throughout the review process. We truly appreciate the time and effort you devoted to coordinating the reviews.
>
> We have now uploaded the camera-ready version of the paper. Please let us know if there are any issues or additional changes needed for approval; we would be very happy to address them promptly.
>
> Thank you again for your service and support!
>
> Best regards,
>
> Authors of Submission 5665